



# Cancellation of cloud shadow effects in the absorbing aerosol index retrieval algorithm of TROPOMI

Victor J. H. Trees [1,2], Ping Wang [1], Piet Stammes [1], Lieuwe G. Tilstra [1], David P. Donovan [1,2], and A. Pier Siebesma [2]

[1]Research & Development Satellite Observations, Royal Netherlands Meteorological Institute (KNMI), Utrechtseweg 297, 3731 GA, De Bilt, the Netherlands
[2]Department of Geoscience & Remote Sensing, Delft University of Technology, Stevinweg 1, 2628 CN, Delft, the Netherlands

**Correspondence:** Victor Trees (victor.trees@knmi.nl)

**Abstract.**

Cloud shadows can be detected in the radiance measurements of the TROPOMI instrument on board the Sentinel-5P satellite due to its high spatial resolution, and could possibly affect its air quality products. The cloud shadow induced signatures are, however, not always apparent and may depend on various cloud and scene parameters. Hence, the quantification of the cloud shadow impact requires the analysis of large data sets. Here we use the cloud shadow detection algorithm DARCLOS to detect cloud shadow pixels in the TROPOMI absorbing aerosol index (AAI) product over Europe during 8 months. For every shadow pixel, we automatically select cloud- and shadow-free neighbour pixels, in order to estimate the cloud shadow induced signature. In addition, we simulate the measured cloud shadow impact on the AAI with our newly developed 3D radiative transfer algorithm MONKI. Both the measurements and simulations show that the average cloud shadow impact on the AAI is close to zero (0.06 and 0.16, respectively). However, the top-of-atmosphere reflectance ratio between 340 and 380 nm, which is used to compute the AAI, is significantly increased in 95% of the shadow pixels. So, cloud shadows are bluer than surrounding non-shadow pixels. Our simulations explain that the traditional AAI formula intrinsically already corrects for this cloud shadow effect, via the lower retrieved scene albedo. This cancellation of cloud shadow signatures is not always perfect, sometimes yielding second order low and high biases in the AAI which we also successfully reproduce with our simulations. We show that the magnitude of those second order cloud shadow effects depends on various cloud parameters which are difficult to determine for the shadows measured with TROPOMI. We conclude that a potential cloud shadow correction strategy for the TROPOMI AAI would therefore be complicated if not unnecessary.

## 1 Introduction

The TROPOspheric Monitoring Instrument (TROPOMI) is a spectrometer onboard the Sentinel-5 Precursor (S5P) satellite in low Earth orbit, launched on October 13, 2017 (Veefkind et al., 2012). TROPOMI provides daily global maps of trace gases, aerosols and clouds, derived from the spectrum of sunlight reflected by the Earth. The spatial resolution TROPOMI of $5.6 \times 3.6$ km$^2$ in the nadir-viewing direction is very high compared to its predecessors, such as OMI with $24 \times 13$ km$^2$ (Levelt et al.,





2006), GOME-2 with $80 \times 40$ km$^2$ (Munro et al., 2016) and SCIAMACHY with $60 \times 30$ km$^2$ (Bovensmann et al., 1999). Because of this unprecedented spatial resolution and its high data quality, TROPOMI is able to observe local NO$_2$ emission
sources such as power plants (Beirle et al., 2019), gas compressor stations (van der A et al., 2020), and cities (Lorente et al., 2019), CH$_4$ leakage from oil/gas fields (Pandey et al., 2019; Varon et al., 2019; Schneising et al., 2020), volcanic SO$_2$ plumes (Theys et al., 2019) and NO$_2$ trails along ship tracks (Georgoulias et al., 2020).

TROPOMI also effectively tracks aerosols that absorb light in the UV part of the spectrum, such as desert dust, volcanic ash, and smoke from biomass burning, by providing the Absorbing Aerosol Index (AAI) in every pixel of each orbit (Stein Zweers
et al., 2018; de Graaf et al., 2005). Unlike most other aerosol retrieval products, the AAI can also successfully be derived above clouds and bright surfaces. In addition, the AAI is an important input for the retrieval algorithms of other TROPOMI products. For example, the pixel selection for the aerosol layer height (ALH) (Sanders et al., 2015; Nanda et al., 2019) and aerosol optical thickness (AOT) (de Graaf, 2022) retrievals of TROPOMI is based on the AAI. Hence, AAI features that are not related to absorbing aerosols, for example caused by the ocean glint and clouds at specific scattering geometries (Kooreman et al., 2020),
may be undesired for those retrievals.

The effect of clouds on the AAI in cloudy pixels has been studied before using data of SCIAMACHY (Penning de Vries et al., 2009; Penning de Vries and Wagner, 2011), OMI (Torres et al., 2018; Jethva et al., 2018) and TROPOMI (Kooreman et al., 2020). Besides cloud signatures in cloudy pixels, clouds can also leave signatures in cloud-free adjacent pixels, for example in the form of cloud shadows. Contrary to the large pixel sizes of its predecessors, the small pixel size of TROPOMI sometimes
causes one or several pixels to be fully covered by a single cloud shadow, particularly for high clouds at large viewing and/or solar zenith angles (Trees et al., 2022). Those three-dimensional radiative transfer effects are not yet taken into account in the current AAI retrieval algorithm, and their influence on the AAI has not yet been investigated. The natural horizontal variation of the AAI complicates the quantification of the cloud shadow induced AAI signatures. Recently, we developed an accurate and fast cloud shadow detection algorithm for TROPOMI, called DARCLOS (Trees et al., 2022), which allows for a statistical
analysis of the cloud shadow effect on the AAI in large data sets.

In this paper, we present a statistical analysis of the cloud shadow effect on the measured TROPOMI AAI for all pixels above Europe during 8 months. We use the cloud shadow detection algorithm DARCLOS to detect the cloud shadow pixels, and we select cloud- and shadow-free neighbour pixels for comparison to the non-shadow state. In addition, we simulate the measured cloud shadow effect on the AAI for various scenes using our 3D radiative transfer code MONKI, recently developed
by us at KNMI. Using our simulations, we explain the measured cloud shadow effects on the AAI. Finally, we discuss the implications of our findings for the TROPOMI AAI product.

This paper is structured as follows. In Sect. 2, we describe the methods we used to measure and to simulate cloud shadow effects on the Absorbing Aerosol Index product of TROPOMI. In Sect. 3., we show the results of those measured and simulated cloud shadow effects. In Sect. 4, we discuss the implications of our results and state the most important conclusions of this
paper.



## 2 Method

In this section, we first give a brief description of TROPOMI (Sect. 2.1), the Absorbing Aerosol Index product (Sect. 2.2) and the data set we selected (Sect. 2.3). Then, we explain the employed methods to detect cloud shadow pixels (Sect. 2.4) and their shadow-free neighbour pixels (Sect. 2.5). Finally, we describe our model to simulate cloud shadow effects (Sect. 2.6).


### 2.1 Description of TROPOMI

The TROPOspheric Monitoring Instrument (TROPOMI) was launched on October 13, 2017, as the only instrument onboard the Sentinel-5 Precursor (S5P) satellite (Veefkind et al., 2012). Operating in a near-polar, Sun-synchronous orbit at an average altitude of 824 km above the Earth's surface, TROPOMI completes an orbit approximately every 101 minutes. TROPOMI is a nadir-looking instrument. During its ascending node, it collects measurements every 1.08 seconds in a 2600 km swath width, 65 providing a daily global coverage. The local equator crossing time of TROPOMI is 13:30 LT. TROPOMI initially featured a footprint size of $7.2 \times 3.6$ km$^2$ in the nadir viewing direction, which was later adjusted to $5.6 \times 3.6$ km$^2$ on August 6, 2018 (Ludewig et al., 2020).

TROPOMI is a spectrometer continuously measuring the Earth radiance, $I_\lambda$, at wavelengths $\lambda$ and in units of W m$^{-2}$ nm$^{-1}$ sr$^{-1}$, and the extraterrestrial solar irradiance perpendicular to the beam, $E_{0\lambda}$, in units of W m$^{-2}$ nm$^{-1}$ daily, to derive the 70 measured local top-of-atmosphere (TOA) reflectance $R_\lambda^{\mathrm{meas}}$:

$$R_\lambda^{\mathrm{meas}}(\mu, \mu_0, \phi, \phi_0) = \frac{\pi I_\lambda(\mu, \mu_0, \phi, \phi_0)}{\mu_0 E_{0\lambda}}, \tag{1}$$

where $\mu = \cos\theta$ and $\mu_0 = \cos\theta_0$, and with $\theta$, $\theta_0$, $\varphi$ and $\varphi_0$ the viewing zenith, solar zenith, viewing azimuth and solar azimuth angles, respectively. From $R_\lambda^{\mathrm{meas}}$, properties of the local Earth's atmosphere and surface can be retrieved. Covering wavelengths in the ultraviolet-visible (UV-VIS, 267–499 nm), near-infrared (NIR, 661–786 nm), and shortwave infrared (SWIR, 2300–2389 75 nm) with high spectral resolution, TROPOMI globally and daily retrieves the concentrations of trace gases (NO$_2$, O$_3$, CH$_4$, CO, and SO$_2$) and properties of aerosols and clouds with unprecedented accuracy.

### 2.2 Absorbing Aerosol Index (AAI)

The air quality product that we analyze is the TROPOMI level 2 Absorbing Aerosol Index (AAI). The AAI is retrieved from the measured and calculated TOA reflectances at 340 and 380 nm as follows (see Torres et al., 1998; de Graaf et al., 2005; 80 Stein Zweers et al., 2018):

$$\mathrm{AAI} = -100 \cdot \left[ \log_{10} \left( \frac{R_{340}}{R_{380}} \right)^{\mathrm{meas}} - \log_{10} \left( \frac{R_{340}}{R_{380}} \right)^{\mathrm{calc}} \right], \tag{2}$$

where 'meas' and 'calc' indicate the measured (Eq. (1)) and calculated TOA reflectances, respectively. The calculated TOA reflectances are for a clear-sky atmosphere above a Lambertian (i.e., isotropically reflecting and fully depolarizing) surface,




and were obtained using the formula of Chandrasekhar (1960):

$$R_\lambda^{\mathrm{calc}}(\mu, \mu_0, \phi - \phi_0) = R_\lambda^0(\mu, \mu_0, \phi - \phi_0) + \frac{A_{\mathrm{s}} T_\lambda(\mu, \mu_0)}{1 - A_{\mathrm{s}} s_\lambda^\star}. \tag{3}$$

In Eq. (3), $R^0$ is the path reflectance, which represents the contribution to the TOA reflectance of the clear-sky atmosphere bounded below by a black surface. The second term in Eq. (3) represents the effect of the surface on the TOA reflectance. It contains the Lambertian surface albedo $A_{\mathrm{s}}$, the total two-way transmittance of the atmosphere $T$, and the spherical albedo $s^\star$ of the atmosphere for illumination from below. Quantities $R^0$, $T$, and $s^\star$ are computed with the 'Doubling-Adding KNMI' (DAK) radiative transfer code (de Haan et al., 1987; Stammes, 2001). This computation accounts for the effects of single and multiple Rayleigh scattering and the absorption of sunlight by molecules within a pseudo-spherical atmosphere, fully taking into account the polarization of light.

The Lambertian surface albedo, $A_{\mathrm{s}}$, in Eq. (3) is retrieved at $\lambda = 380$ nm assuming that $R_{380}^{\mathrm{calc}}(A_{\mathrm{s}}) = R_{380}^{\mathrm{meas}}$. The value of $A_{\mathrm{s}}$ which satisfies this assumption is known as the 'scene albedo' or the 'scene Lambertian equivalent reflectance (scene LER)', and denoted in this paper as $A_{\mathrm{scene}}$. From Eq. (3) it then follows that

$$A_{\mathrm{scene}} = \frac{R_{380}^{\mathrm{meas}} - R_{380}^0}{T_{380}(\mu, \mu_0) + s_{380}^\star (R_{380}^{\mathrm{meas}} - R_{380}^0)}. \tag{4}$$

$A_{\mathrm{scene}}$ is assumed wavelength independent, allowing for the computation of $R_{340}^{\mathrm{calc}}$ using Eq. (3) but with $\lambda = 340$ nm. Finally, the AAI is computed using Eq. (2).

In the absence of aerosols and clouds, the AAI is, in theory, ideally equal to zero. Due to spectrally varying instrument imperfections, the AAI of TROPOMI has an offset of $\sim -2$. The AAI tends to increase in the presence of absorbing aerosols and can also identify aerosols that are located above clouds. We refer to Herman et al. (1997), Torres et al. (1998), de Graaf et al. (2005), Penning de Vries et al. (2009) and Kooreman et al. (2020) for more details about the sensitivity of the AAI to aerosols, surfaces and clouds. For this research about cloud shadow effects, it should be noted that dark pixels (low $R_{380}^{\mathrm{meas}}$) give low $A_{\mathrm{scene}}$ (Eq. (4)) resulting in relatively small contributions of the surface to $R_{340}^{\mathrm{calc}}$ and $R_{380}^{\mathrm{calc}}$ (Eq. (3)).

## 2.3 Selected data set

We analyzed 8 months of TROPOMI AAI data (processor version 1.3.0), from 1 November 2020 to 30 June 2021. The selected area ranges from 34 °S to 61 °N latitude, and from 11 °W to 40 °E longitude, as shown in the left figure of Fig. 1. This area covers all capitals of Europe (except for Reykjavik), Ankara, Moscow, and some Northern African cities such as Tanger, Algiers and Tunis. As shown in Figure 1, the grids of 3 orbits intersect on 11 November 2020 the selected area over Europe, and partly overlap each other. For each day in the data set, we use all pixels that fall in the selected area, on average resulting in 511616 pixels per day available for our analysis. Days with missing data (2021-05-23, 2021-05-20 and 2021-06-29), inconsistent ground pixel grids of the AAI and NO$_2$ products (2021-06-24) and solar eclipse (2021-06-10) were removed from the data set.





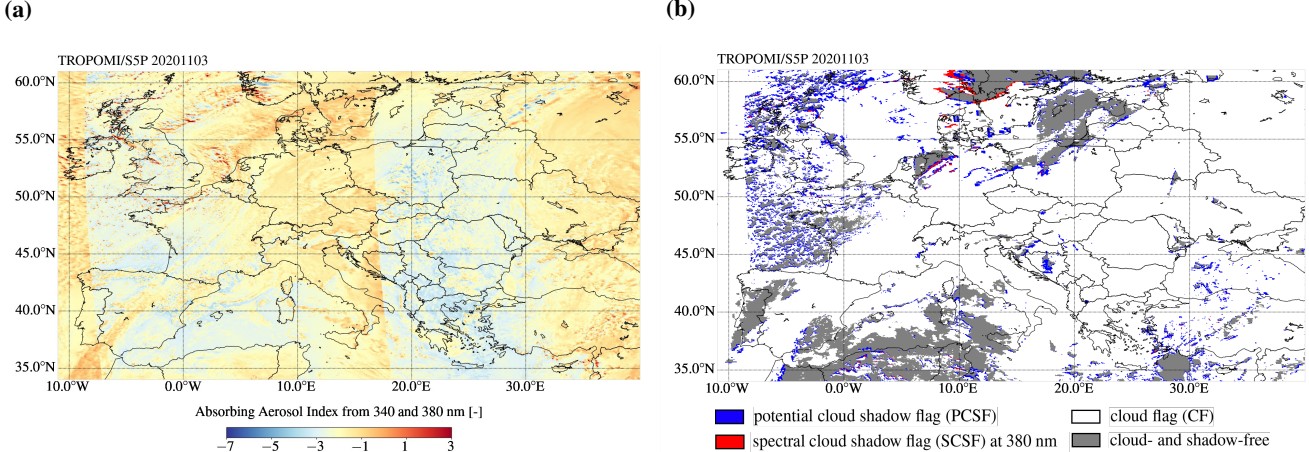

**Figure 1.** Example of the AAI (left) and cloud shadow flags (right) in three partly overlapping TROPOMI orbit swaths covering the selected area for this research, here on 3 November 2020. In the right figure, the blue pixels contain a potential cloud shadow flag (PCSF), the red pixels contain (in addition) a spectral cloud shadow flag (SCSF), the white pixels contain a cloud flag (CF) and the grey pixels are cloud- and shadow-free.

## 2.4 Cloud shadow detection

The flagging of pixels affected by cloud shadows was performed with the cloud shadow detection algorithm DARCLOS, recently developed for TROPOMI at KNMI (see Trees et al., 2022).

In DARCLOS, first, cloud pixels are identified using a threshold on the already available effective cloud fraction in the TROPOMI $NO_2$ product (van Geffen et al., 2021), after which cloud flags (CFs) are raised. Then, potential cloud shadow flags (PCSFs) are raised, indicating TROPOMI ground pixels that are potentially affected by cloud shadows. The PCSFs are determined using a geometrical calculation of the shadow location based on the cloud height from the TROPOMI cloud product FRESCO (Koelemeijer et al., 2001; Wang et al., 2008) and illumination and viewing geometries. The PCSFs generally overestimate the true visible cloud shadow area but minimize the omission of pixels affected by cloud shadows.

After the PCSFs are raised, DARCLOS raises spectral cloud shadow flags (SCSFs). The SCSFs are a subset of the PCSFs, based on a threshold on the contrast $\Gamma$ between the retrieved scene albedo, $A_{\text{scene}}$ (see Eq. 4), and the expected surface albedo from a climatology, $A_{\text{DLER}}$. An SCSF is raised for a pixel if (see Eqs. 11 and 17 of Trees et al., 2022)

$$\Gamma(\lambda) < -15\%, \tag{5}$$

where

$$\Gamma(\lambda) = \frac{A_{\text{scene}}(\lambda) - A_{\text{DLER}}(\lambda)}{A_{\text{DLER}}(\lambda)} \times 100\%. \tag{6}$$

The variable $A_{\text{DLER}}$ is also known as the directionally dependent Lambertian-equivalent reflectivity or DLER (see Tilstra et al., 2023). For this research about cloud shadow effects on the AAI retrieved at 340 and 380 nm, we employ $\lambda = 380$ nm. The





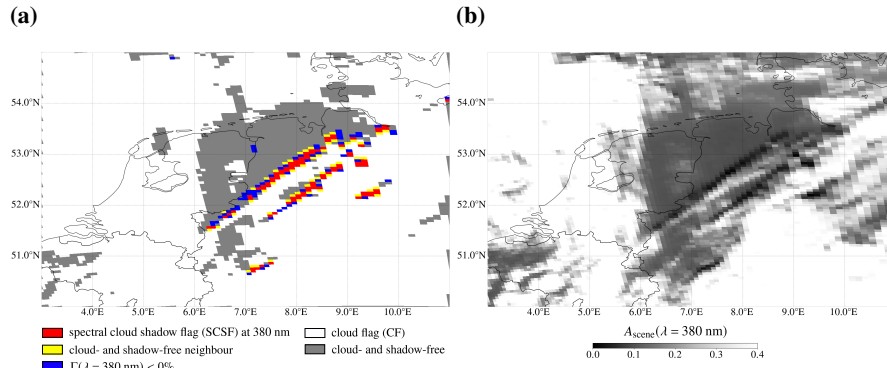

**Figure 2.** TROPOMI pixels with spectral cloud shadow flags (SCSFs) at $\lambda = 380$ nm in red, with cloud flags (CFs) in white, first cloud- and shadow-free neighbour pixels in yellow, possibly shadow affected pixels (Eq. (7)) in blue, and remaining TROPOMI pixels in grey, on 3 November 2020 above the Netherlands, Belgium and North-West Germany (Fig. 2a). The corresponding scene albedo at 380 nm, $A_{\mathrm{scene}}(\lambda = 380$ nm), derived by TROPOMI (Fig. 2b).

SCSFs are a better estimate of the cloud shadows than the PCSFs. As an example, the right figure in Fig. 1 shows the SCSFs indicated in red, the PCSFs indicated in blue and the CFs indicated in white, in three TROPOMI orbits covering the area of our case study, at 3 November 2020 which is one of the days in our data set. For more details about the cloud shadow flagging with DARCLOS, we refer to Trees et al. (2022).

## 2.5  Selecting cloud- and shadow-free neighbours

In order to be able to quantify the cloud shadow effect on the AAI in a shadow pixel (i.e., for which a SCSF was raised), we identify cloud- and shadow-free reference pixels in the proximity of the shadow pixel and assume that they represent the hypothetical non-shadow state of the shadow pixel, that is, as if the shadow pixel would not be affected by cloud shadow. In what follows, we call those reference pixels the neighbour pixels. We distinguish between first and second neighbour pixels, for the closest and second-closest neighbour pixels respectively. The second neighbour pixels are used for comparison to the

first neighbour pixels as a control case, since both the first and second neighbour pixels should not be affected by cloud shadow.

First, for each shadow pixel, we define a search area with potential neighbour pixels of two TROPOMI pixels around the shadow pixel. That is, a neighbour pixel cannot be located more than two scanlines, or more than two pixel rows, away from the shadow pixel. Because we require the neighbour pixels to be cloud- and shadow-free, the cloud pixels and shadow pixels are removed from the search area. Some pixels that are darker than expected are possibly (partly) affected by cloud shadows

but not severely enough to raise a SCSF by DARCLOS (see Eqs. 5 and 6). Because we do not trust them as shadow-free pixels, they are removed from the search area when

$$\Gamma(\lambda = 380 \text{ nm}) < 0\%. \tag{7}$$

For each left-over potential neighbour pixel in the search area, we compute the distance in latitude-longitude space from the center of the potential neighbour to the center of the shadow pixel. The left-over potential neighbour pixel with the closest

distance to the shadow pixel is selected as the first neighbour pixel for this shadow pixel. Similarly, we define the second





neighbour pixel as the left-over potential neighbour pixel with the second-closest distance to the shadow pixel. Only if both a first and second neighbour pixel can be determined, the shadow pixel is considered in our analysis.

Figure 2a shows an example of the SCSFs at $\lambda = 380$ nm indicated in red and the first neighbour pixels indicated in yellow, for 3 November 2020 above North-West Germany. In this scene, cloud shadows are found northward of the clouds between

$6.0°$ and $10°$E longitude. The pixels that could not be selected as a neighbour because $\Gamma < 0\%$ (see Eq. (7)) are indicated in blue. Note that there are less neighbour pixels than raised SCSFs, because (1) some shadow pixels do not have at least two cloud- and shadow-free pixels with $\Gamma \geq 0\%$ in their search area and (2) neighbours can be recycled for multiple shadow pixels.

Figure 2b shows $A_{\text{scene}}$ retrieved by TROPOMI at $\lambda = 380$ nm, for the same scene as in Fig. 2a. From visual comparison of Figs. 2a and 2b, it may be observed that the SCSFs are indeed located at pixels where $A_{\text{scene}}$ is lower than at surrounding pixels

along cloud edges, which may be interpreted as cloud shadows. The first neighbour pixels are indeed not located where there are clouds (i.e., pixels for which $A_{\text{scene}} \gtrsim 0.3$ in Fig. 2b) or where possibly cloud shadow darkening occurs.

## 2.6 Simulating the cloud shadow effect

Three-dimensional radiative transfer simulations are required for the explanation of cloud shadow effects on the AAI as found in the observations. In this research, we use the three-dimensional radiative transfer code MONKI (Monte Carlo KNMI), that

we recently developed at KNMI. MONKI computes the TOA reflectance of an atmosphere-surface system defined in a 3D Cartesian grid in a horizontally cyclic domain, using a forward Monte Carlo technique (see, e.g., Marshak and Davis, 2005), and fully takes into account linear and circular polarization of light for all orders of scattering. The simulated photon packets travelling through the grid cells of the atmosphere-surface system are scattered by the atmospheric gas through (anisotropic) Rayleigh scattering (Hansen and Travis, 1974), and by cloud droplets through Mie scattering (de Rooij and van der Stap, 1984)

if the grid cell is cloudy. Absorption of the light by the gas and by cloud droplets is taken into account. The surface reflects Lambertian (i.e., isotropic and fully depolarizing), with a specified surface albedo. Instead of collecting the reflected photon packets at TOA in the very small solid angle subtended by the satellite, MONKI uses the more efficient 'local estimation method' (Marchuk et al., 1980; Marshak and Davis, 2005), commonly used in Monte Carlo radiative transfer algorithms (see e.g. Spada et al., 2006; Mayer, 2009; Deutschmann et al., 2011), in which at each scattering event the contribution to the re-

flectance is computed as the probability that the photon is being scattered towards the satellite. The TOA reflectance of MONKI has been compared to the DAK (Doubling Adding KNMI) polarized radiative transfer code (de Haan et al., 1987; Stammes, 2001), for plane-parallel and horizontally homogeneous cloudy and cloud-free scenes, and shows an excellent agreement.

For this research, we use 50 by 50 grid cells in the horizontal directions of the 200 by 200 km$^2$ cyclic domain and 33 grid cells in the vertical direction ranging from 0 km to 100 km. Ozone is the only absorbing gas in our model, with absorption cross-

sections taken from Bass and Paur (1985). The pressure, temperature and ozone volume mixing ratios solely depend on altitude according to the standard Mid-Latitude Summer profile (Anderson, 1986). We assume that the pressure-temperature ratio decreases exponentially with height within each grid cell (see Stam et al., 2000, for the calculation of the gaseous absorption and scattering optical thicknesses). For our analysis, we vary the cloud optical thickness, cloud height, cloud horizontal dimensions,



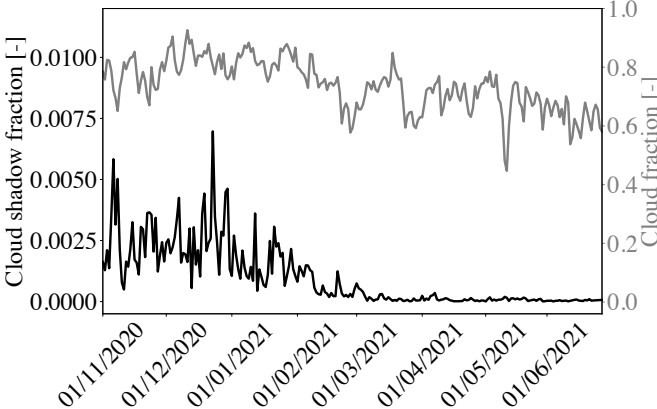

**Figure 3.** Daily cloud fraction (grey curve) and cloud shadow fraction (black curve) in the selected area over Europa from 1 November 2020 until 30 June 2021.

solar and viewing zenith and azimuth angles, and surface albedo, resulting in various sizes and shades of cloud shadows in the

atmosphere and cast on the surface.

## 3   Results

In this section, we present the results of the determined cloud shadow fraction in our selected data set (Sect. 3.1), and of the measured (Sect. 3.2) and simulated (Sect. 3.3) cloud shadow effect on the TROPOMI AAI.

### 3.1   Cloud shadow fraction

Figure 3 shows the cloud fraction and cloud shadow fraction in the selected area over Europe, from 1 November 2020 to 30 June 2021. The cloud fraction and cloud shadow fraction were defined as the fraction of pixels with a raised CF and SCSF, respectively. The cloud shadow fraction was relatively large from 1 November to 31 January with an average daily mean value of 0.002, which corresponds to 1124 cloud shadow pixels on average per day. After January, the cloud shadow fraction decreases. From 1 March to 30 June, the average daily mean cloud shadow fraction was $7.4 \cdot 10^{-5}$, corresponding to 38 cloud

shadow pixels on average per day. We note that changes in the trend of the daily mean cloud fraction are much less apparent. The higher cloud shadow fraction in the winter months than in the spring and summer months can be explained by the larger solar zenith angles, resulting in (1) longer shadow extents and (2) increased shadow darkness due to the longer slant path length of the incoming direct light through the clouds.

### 3.2   Measured cloud shadow effects on the AAI

Here we present the results of the measured cloud shadow effect on the TROPOMI AAI. That is, we compare the AAI in the shadow pixels of our data set with the AAI in their first cloud- and shadow-free neighbour pixels. To show the natural variation





irrespective of cloud shadows, we also compare the AAI in the first and second neighbour pixels. In addition, we analyze the results for the measured TOA reflectance ratio $(R_{340}/R_{380})^{\mathrm{meas}}$ and calculated TOA reflectance ratio $(R_{340}/R_{380})^{\mathrm{calc}}$ which determine the AAI (Eq. (2)), and the retrieved scene albedo $A_{\mathrm{scene}}$ which determines $(R_{340}/R_{380})^{\mathrm{calc}}$ via $R_{340}^{\mathrm{calc}}$ (Eq. (3)).

**3.2.1 First order cloud shadow effect**

Figure 4 shows the AAI (top row) in the first neighbour pixels compared in a scatter plot to the second neighbour pixels (first column), and to the shadow pixels (second column). In both cases, the scatter plots show a high positive correlation ($r = 0.82$ and $0.81$, respectively). The AAI in the shadow pixels is not consistently larger or smaller than the AAI in the first neighbour pixels: 55% of the shadow pixels show a larger AAI compared to their first neighbour pixels. This inconsistency is observed
throughout the complete time span the data set, which is clear from the daily mean AAI time series in the third column of Fig. 4: the daily mean AAI in the shadow pixels is higher on some days and lower on other days compared to their first neighbour pixels. Note that the number of shadow pixels is significantly smaller in the spring months in the second half of the data set, as seen in Fig. 3, which increases the uncertainty of the daily mean AAI. The fourth column of Fig. 4 shows that the histograms of the AAI difference between the shadow pixels and first neighbour pixels is approximately centered around 0 (the mean AAI
difference is $0.055 \pm 0.002$), and is about an order of magnitude smaller than the standard deviation of the data set ($\sigma = 0.316$).

The second row of Fig. 4 shows the measured TOA reflectance ratio $(R_{340}/R_{380})^{\mathrm{meas}}$. Interestingly, although in the previous paragraph we reported no consistent cloud shadow effect on the AAI, the value of $(R_{340}/R_{380})^{\mathrm{meas}}$ in the AAI formula (Eq. (2)) is consistently higher in the shadow pixels than in their first neighbour pixels: 95% of the shadow pixels show a larger $(R_{340}/R_{380})^{\mathrm{meas}}$ compared to their first neighbour pixels. This higher reflectance ratio in the shadow pixels is observed on
all days (figure in the third column and second row of Fig. 4) and clearly alters its distribution (figure in the fourth column and second row of Fig. 4). The difference of the mean $(R_{340}/R_{380})^{\mathrm{meas}}$ in the shadow pixels with respect to their neighbours is $0.036$, which is larger than the standard deviation $\sigma = 0.026$. Those results imply that the measured TROPOMI UV TOA reflectances were consistently 'more blue' in the shadow pixels than in the neighbour pixels.

The missing cloud shadow effect on the AAI, while TROPOMI consistently measured higher values for $(R_{340}/R_{380})^{\mathrm{meas}}$
in the cloud shadows, can be explained by the behaviour of the calculated reflectance ratio $(R_{340}/R_{380})^{\mathrm{calc}}$. Indeed, as shown in the third row of Fig. 4, $(R_{340}/R_{380})^{\mathrm{calc}}$ is also elevated in the shadow pixels, which happens to be similar to the increase of $(R_{340}/R_{380})^{\mathrm{meas}}$. This increase of $(R_{340}/R_{380})^{\mathrm{calc}}$ is caused by the lower retrieved scene albedo $A_{\mathrm{scene}}$ in the shadow pixels, as a result of the lower measured reflectance $R_{380}^{\mathrm{meas}}$ in the cloud shadows (see Eq. (4)). With lower $A_{\mathrm{scene}}$, the contribution of the (spectrally flat) Lambertian surface in the DAK model decreases, which increases the 'blueness' of the calculated TOA
reflectances (Eq. (3)) and thus increases $(R_{340}/R_{380})^{\mathrm{calc}}$. A similar effect on $(R_{340}/R_{380})^{\mathrm{calc}}$ can be found during solar eclipses (Trees et al., 2021). However, in contrast to cloud shadows, the lunar shadow is imposed from outside the Earth system, such that the light paths in principle do not change and $(R_{340}/R_{380})^{\mathrm{meas}}$ is not altered, resulting in a strong increase of the AAI during solar eclipses. In cloud shadows, simultaneous increases of $(R_{340}/R_{380})^{\mathrm{meas}}$ and $(R_{340}/R_{380})^{\mathrm{calc}}$ lead, to the first order, to cancellations of cloud shadow effects in the AAI through Eq. (2). The explanation of the missing first order
cloud shadow effect is explained in more detail using our simulation results in Section 3.3.1.





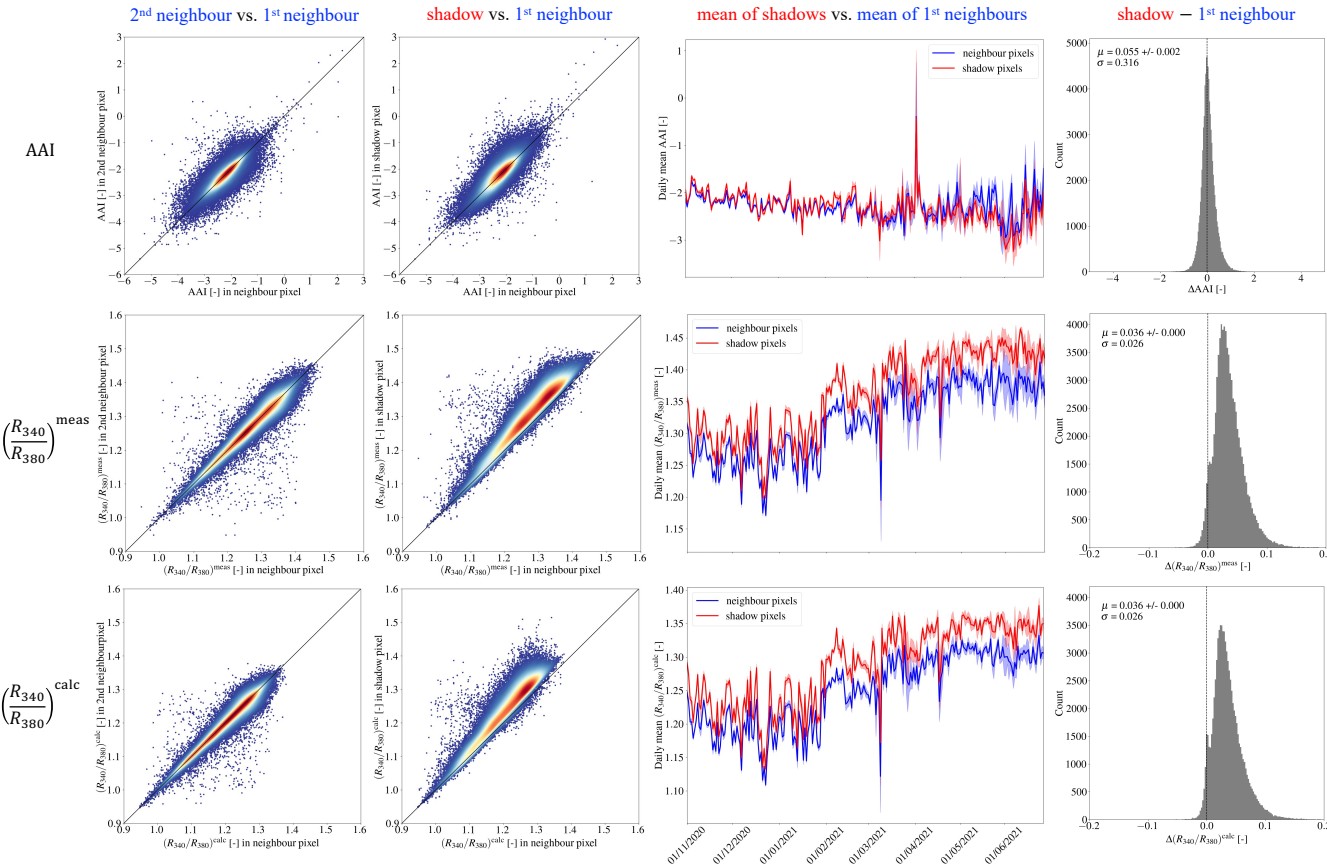

**Figure 4.** Comparison of the values in the second and first neighbour pixels (first column), the cloud shadow pixels and the first neighbour pixels (second column), the daily mean values of the cloud shadow pixels and the first neighbour pixels (third column), and the histograms of their differences (shadow - first neighbour, fourth column), for the TROPOMI AAI (first row), the measured TOA reflectance ratio between 340 and 380 nm (second row), and the calculated TOA reflectance ratio between 340 and 380 nm in the AAI retrieval algorithm (third row).

The cancellation of the cloud shadow effect on the AAI is also apparent in the AAI map of a single cloud shadow case. Fig. 5 shows maps of the AAI (first column), $(R_{340}/R_{380})^{\mathrm{meas}}$ (second column) and $(R_{340}/R_{380})^{\mathrm{calc}}$ (third column) over the Netherlands, Belgium and North-West Germany on 3 November 2020. From Fig. 2, it was known that cloud shadows were present in this scene northward of the clouds between 6.0° and 10°E longitude. Indeed, those cloud shadows appear to have increased $(R_{340}/R_{380})^{\mathrm{meas}}$ and $(R_{340}/R_{380})^{\mathrm{calc}}$, which is clear from the darker blue shade compared to their cloud- and shadow-free surroundings. In the AAI map, the cloud shadows can hardly be distinguished from their surroundings.




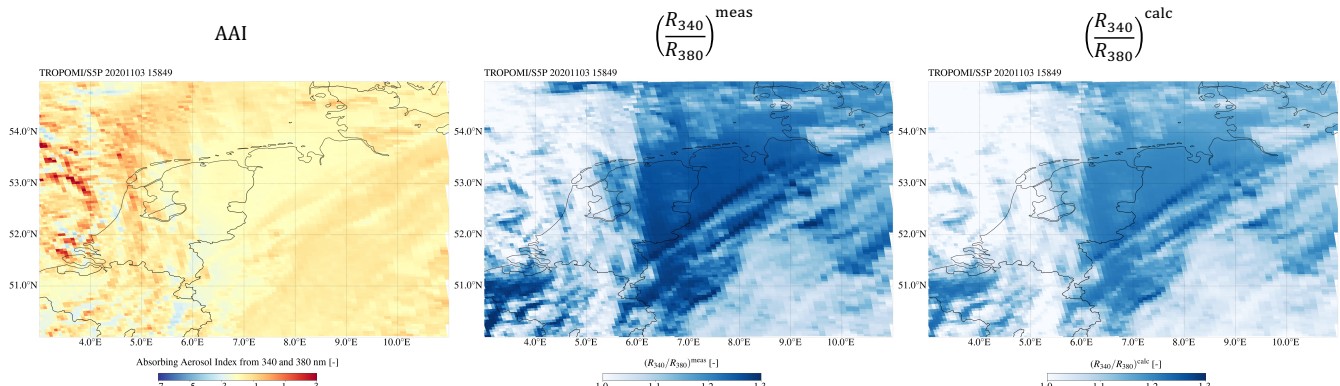

**Figure 5.** Examples of the TROPOMI AAI (first column), the measured TOA reflectance ratio between 340 and 380 nm (second column) and the calculated TOA reflectance ratio between 340 and 380 nm in the TROPOMI AAI retrieval (third column), for 3 November 2020 over the Netherlands and Germany.

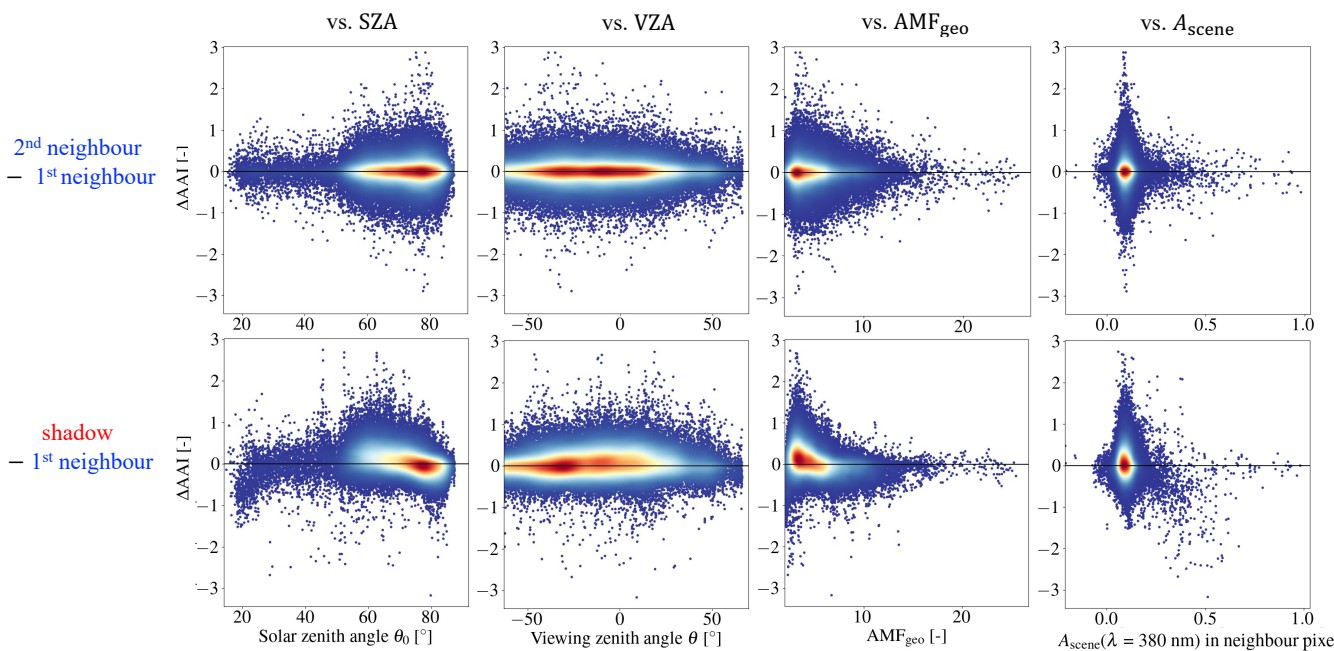

**Figure 6.** Similar AAI data as in Fig. 4, but now plotted as the differences between the second and first neighbour pixels (first row) and between the cloud shadow pixels and first neighbour pixels (second row), as functions of solar zenith angle (first column), viewing zenith angle (second column), geometric air mass factor (third column) and retrieved scene albedo at 380 nm in the first neighbour pixel (fourth column).



### 3.2.2 Second order cloud shadow effects

Although $(R_{340}/R_{380})^{\mathrm{meas}}$ and $(R_{340}/R_{380})^{\mathrm{calc}}$ are more blue in the shadow than their surroundings in almost all cases, we found that the cancellation of cloud shadow effects in the measured AAI as discussed in the previous Section is not always

perfect. We investigated the dependency of the second order cloud shadow effect on the AAI to physical parameters, and found a slight dependency on the illumination and viewing geometries, and the surface albedo. We call those the second-order cloud shadow effects.

Figure 6 shows the AAI difference between the shadow and first neighbour pixels (bottom row) versus the solar zenith angle $\theta_0$ (first column), viewing zenith angle $\theta$ (second column), geometric air mass factor $\mathrm{AMF}_{\mathrm{geo}} = 1/\mu + 1/\mu_0$ (third column),

and scene albedo $A_{\mathrm{scene}}$ of the first neighbour pixel (fourth column). Also, we present the differences between the second and first neighbour pixels (top row), which represent the natural AAI variation irrespective of cloud shadows. From the results in the bottom row, it can be concluded that cloud shadows tend to increase the AAI slightly for decreasing $\theta_0$ from $80°$ to $\sim 50°$ (the mean $\Delta$AAI is 0.18 in the shadow case between $\theta_0 = 50°$ and $70°$), and for small $\theta$ (the mean $\Delta$AAI is 0.09 in the shadow case for $|\theta| < 30°$). This dependency, however, seems to be more apparent when combining $\theta$ and $\theta_0$ in the geometric air mass

factor: smaller $\mathrm{AMF}_{\mathrm{geo}}$ give slightly increased AAI (the mean $\Delta$AAI is 0.14 in the shadow case for $\mathrm{AMF}_{\mathrm{geo}} < 5$). Another dependency can be measured in $A_{\mathrm{scene}}$: bright surfaces ($A_{\mathrm{scene}} \gtrsim 0.2$) tend to decrease the AAI. However, for most pixels in our data set $A_{\mathrm{scene}}$ equals approximately 0.1, for which this dependency does not seem apparent.

### 3.3 Simulated cloud shadow effects on the AAI

Here we present the results of the simulated cloud shadow effect on the TROPOMI AAI using our 3D radiative transfer code

MONKI. We considered a box-shaped cloud with dimensions of 10 x 10 x 1 km$^3$ consisting of spherical droplets with effective radius $r_{\mathrm{eff}}$ of 2.0 $\mu$m, and placed it at three altitudes: at 2, 5 and 10 km cloud base height $h_{\mathrm{c}}$. Furthermore, we varied the cloud optical thickness ($\tau_{\mathrm{c}} = 1, 5,$ or $10$), the surface albedo ($A_{\mathrm{s}} = 0, 0.1,$ or $0.2$), the viewing zenith angle ($\theta = 0°, 30°, 45°, 60°,$ or $75°$), the solar zenith angle ($\theta_0 = 0°, 30°, 45°, 60°,$ or $75°$), and the azimuth difference ($\phi - \phi_0 = 0°$ or $180°$), resulting in 1350 simulated scenes.

### 3.3.1 First order cloud shadow effect

The first row of Fig. 7 shows an example of the simulated TOA reflectances that would be measured with TROPOMI at 340 and 380 nm, $R_{340}^{\mathrm{meas}}$ and $R_{380}^{\mathrm{meas}}$ respectively, together with their ratio $R_{340}^{\mathrm{meas}}/R_{380}^{\mathrm{meas}}$, for a scene with $h_{\mathrm{c}} = 5$ km, $\tau_{\mathrm{c}} = 10$, $\theta_0 = 75°$, $\theta = 0°$ and $\phi - \phi_0 = 0°$ (i.e., the instrument is nadir-viewing and the Sun is located on the left side of the scene). Here, we assume a black surface ($A_{\mathrm{s}} = 0.0$). The cloud, located at $x = 48 - 68$ km and $y = 28 - 48$ km gives the strongest

signal in both $R_{340}^{\mathrm{meas}}$ and $R_{380}^{\mathrm{meas}}$, due to light multiply scattered by the rather thick cloud towards the satellite instrument, which is approximately wavelength-independent resulting in a white appearance of the cloud ($R_{340}^{\mathrm{meas}}/R_{380}^{\mathrm{meas}} \approx 1$). Outside the cloudy region, the signal is more 'blue' ($R_{340}^{\mathrm{meas}}/R_{380}^{\mathrm{meas}} > 1$), due to the $\lambda^{-4}$ dependence of the Rayleigh scattering optical thickness of the gas. In the cloud shadow, located along the right edge of the cloud, the signal of $R_{340}^{\mathrm{meas}}$ and $R_{380}^{\mathrm{meas}}$ is smallest. Indeed,





the 'blueness' of the cloud shadow is even larger than that of the cloud- and shadow-free region ($R_{340}^{\text{meas}}/R_{380}^{\text{meas}} \gg 1$), which
was also found in the observations by TROPOMI (see Sect. 3.2.1).

In order to explain the 'blue' appearance of cloud shadows as seen from space, we analyze the vertical profiles of the contributions to $R_{340}^{\text{meas}}$, $R_{380}^{\text{meas}}$ and $R_{340}^{\text{meas}}/R_{380}^{\text{meas}}$, in the second row of Figure 7. In the cloud- and shadow-free region, most signal originates from below $\sim 15$ km where the gas pressure (and consequently the Rayleigh scattering optical thickness) is largest. This signal of the background gas is larger at 340 nm than at 380 nm due to the $\lambda^{-4}$ dependence of the Rayleigh scattering
optical thickness of the gas. We note that the 'color' of the background contribution changes from 'blue' ($R_{340}^{\text{meas}}/R_{380}^{\text{meas}} > 1$), through 'white' ($R_{340}^{\text{meas}}/R_{380}^{\text{meas}} = 1$), to 'red' ($R_{340}^{\text{meas}}/R_{380}^{\text{meas}} < 1$), with decreasing altitude, as the blue light has been scattered out of the direct beam that is incident on the lowest atmospheric layers. The contribution from the surface is equal to 0, because all the light reaching the surface is absorbed as $A_{\text{s}} = 0.0$ in this example.

Directly below the cloud, from 5 km to the surface, the contribution to both $R_{340}^{\text{meas}}$ and $R_{380}^{\text{meas}}$ is approximately 0, as the
nadir-looking instrument cannot look through the rather thick cloud. Inside the cloud shadow volume, located on the lower right side of the cloud, the contributions are indeed smaller than in the cloud- and shadow-free region but still larger than 0 (see second row of Fig. 7). Contrary to the cloud- and shadow-free contribution at those altitudes, the color of the cloud shadow contribution is blue ($R_{340}^{\text{meas}}/R_{380}^{\text{meas}} > 1$). Comparing the vertical profiles in the cloud- and shadow-free region (e.g., at $x = 26$ km) and through the cloud shadow (e.g., at $x = 74$ km), it can readily be concluded that a vertical integration of the
contribution indeed leads to a relatively blue cloud shadow signal at TOA, thus a higher $R_{340}^{\text{meas}}/R_{380}^{\text{meas}}$ in the cloud shadow than in the cloud- and shadow-free surroundings (see upper right figure in Fig. 7).

We further investigated the nonzero and blue contribution of the cloud shadow volume by separating the vertical profiles of the contributions from single scattering only (third row of Fig. 7) and of multiple scattering only (fourth row of Fig. 7). Note that the sum of those contributions is again the total contribution as shown in the second row of Fig. 7. In the cloud shadow,
there is no contribution from single scattering in our simulation, as shown by the black shades in the contributions to $R_{340}^{\text{meas}}$ and $R_{380}^{\text{meas}}$, and the undefined values in the contribution to $R_{340}^{\text{meas}}/R_{380}^{\text{meas}}$. Apparently, all photons were scattered away from the direct beam passing through the cloud before reaching the cloud base. In multiply scattered light, however, photons can reach the cloud shadow volume after being scattered by the gas in the cloud- and shadow-free region, as illustrated by the nonzero values in the cloud shadow in the fourth row of Fig. 7. Also, it should be noted that, regardless of cloud shadows, the signal
from multiple scattering is more blue than that of single scattering (cf. third and fourth rows of Fig. 7). Thus, because in cloud shadows the singly scattered light is intercepted and the multiply scattered light is left, and because the multiply scattered light is more blue than singly scattered light, the appearance of cloud shadows is relatively blue (second row).

Finally, we computed the AAI that would be retrieved from our simulated TOA reflectances. That is, we used the simulated $R_{340}^{\text{meas}}$ and $R_{380}^{\text{meas}}$ as input for the TROPOMI AAI retrieval algorithm (see Sect. Sect. 2.2). The last row of Fig. 7 shows the
retrieved AAI, $A_{\text{scene}}(380\,\text{nm})$, and $R_{340}^{\text{calc}}/R_{380}^{\text{calc}}$. In the cloud- and shadow-free region the simulated AAI equals 0, since $R_{380}^{\text{meas}}$ and $R_{380}^{\text{calc}}$ are virtually identical due to the excellent agreement between MONKI and DAK for clear-sky scenes. In the cloud shadow, the AAI is also approximately 0 but $A_{\text{scene}}(380\,\text{nm})$ appears dark as a direct result of the lower $R_{380}^{\text{meas}}$ (see Eq. (4)). Note that for this scene with a black surface, the retrieved $A_{\text{scene}}(380\,\text{nm})$ is even negative in the cloud shadow (it should

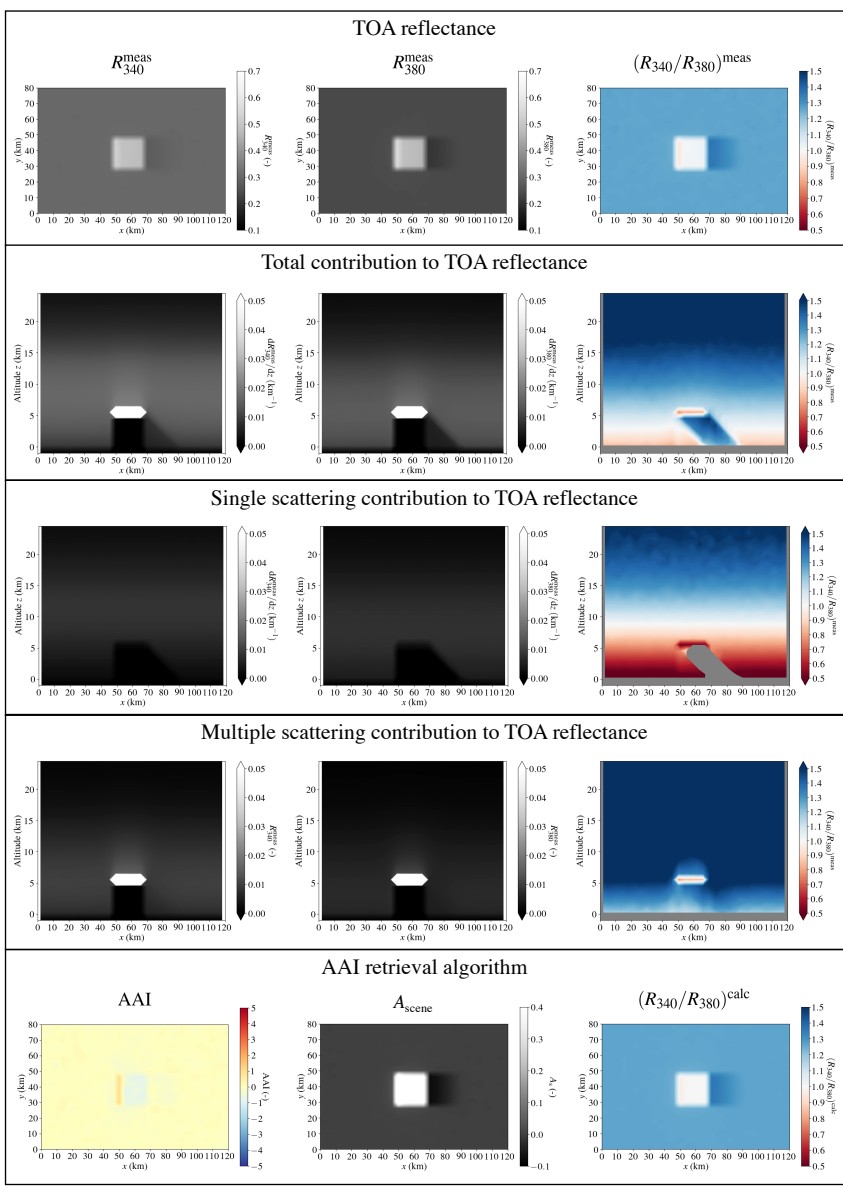

**Figure 7.** Simulations by MONKI of the measured TOA reflectance (top row) and the vertical profiles (in km$^{-1}$) of the contribution to the TOA reflectance for all photons (second row), for single scattering only (third row) and for multiple scattering only (fourth row), at 340 nm (first column), at 380 nm (second column) and for their ratio (third column). The bottom row shows the corresponding AAI (first column), the scene albedo at 380 nm (second column) and the calculated TOA reflectance ratio between 340 and 380 nm using the TROPOMI AAI retrieval algorithm (third column). The vertical profiles were made using the mean values from $y = 30$ to $y = 46$ km. The data point below 0 km altitude represents the contribution of the surface.





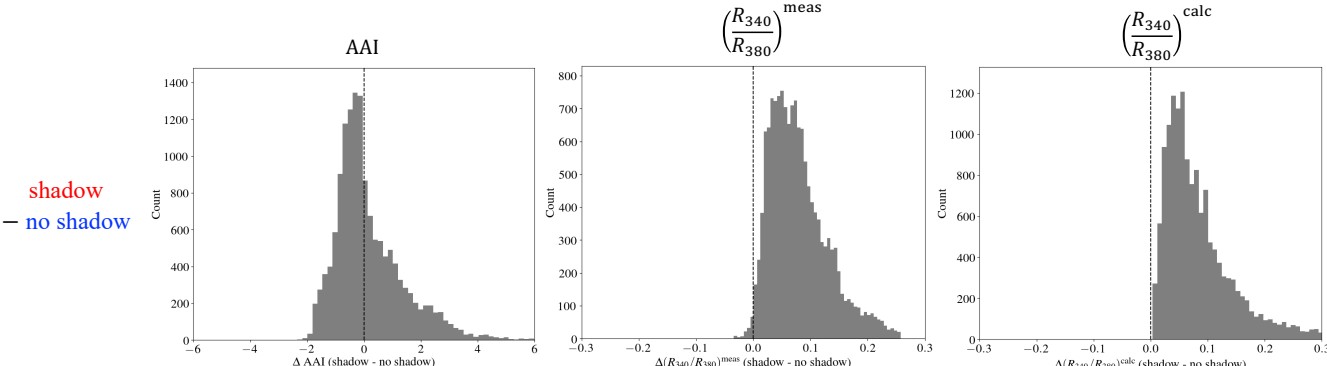

**Figure 8.** Histograms of the differences in the simulated values by MONKI between cloud shadow pixels and cloud- and shadow-free pixels, for the AAI (first column), the measured TOA reflectance ratio between 340 and 380 nm (second column) and the calculated TOA reflectance ratio between 340 and 380 nm using the TROPOMI AAI retrieval algorithm (third column), for all 1350 simulated scenes.

be noted that a negative albedo is non-physical and a result of the AAI algorithm as explained in Sect. 2.2). Consequently, the

calculated (spectrally flat) surface contribution to $R_{340}^{\mathrm{calc}}$ is relatively small in the cloud shadow and is negative, resulting in a relatively large and blue contribution of the path reflectance $R^0$ (see Eq. (3)). Indeed, $R_{340}^{\mathrm{calc}}/R_{380}^{\mathrm{calc}}$ is enhanced in the cloud shadow, as shown in the last row of Fig. 7. Because cloud shadows are more blue in both $R^{\mathrm{calc}}$ and $R^{\mathrm{meas}}$ (note that the figures of $R_{340}^{\mathrm{calc}}/R_{380}^{\mathrm{calc}}$ and $R_{340}^{\mathrm{meas}}/R_{380}^{\mathrm{meas}}$ look approximately identical in Fig. 7), there is no visible cloud shadow signature in the AAI (lower left figure in Fig. 7). In conclusion, the AAI retrieval already automatically corrects for cloud shadows via the

lower retrieved scene albedo, which is in agreement with the first order cloud shadow effect that we found in the TROPOMI observations (Sect. 3.2.1).

We note that at the opposite side of the cloud, the AAI is slightly increased in our simulations (AAI $\sim$1). This increase is found on the cloud itself at $x = 50$ km, where the cloud is directly illuminated by the sun from the side. We speculate that this result demonstrates the positive TROPOMI AAI signatures at the bright side of vertical cloud structures at high latitudes, that

were found by Kooreman et al. (2020) (see their Figs. 1 and 2). Numerical experiments indeed showed that an increase of the cloud vertical extent from 1 km to 5 km further increased this positive AAI signature. Because the scope of our article is the analysis of shadows, we leave the analysis of the bright side of clouds for future research.

In the previous paragraphs of this Section, we discussed the results of one simulated scene. As mentioned in the beginning of this Section, we did those simulations for 1350 scenes. Figure 8 shows the difference between the AAI in the simulated shadow

pixels and the AAI in the cloud- and shadow-free regions of all simulated scenes. In addition, we show those differences for $R_{340}^{\mathrm{meas}}/R_{380}^{\mathrm{meas}}$ and $R_{340}^{\mathrm{calc}}/R_{380}^{\mathrm{calc}}$. For almost all cases (99% and 100% for the measured and calculated values respectively), indeed the cloud shadow pixels are more blue than their cloud- and shadow-free surroundings, and $\Delta R_{340}^{\mathrm{meas}}/R_{380}^{\mathrm{meas}} = 0.079$ and $\Delta R_{340}^{\mathrm{calc}}/R_{380}^{\mathrm{calc}} = 0.086$. The consistent enhanced blueness in both $R^{\mathrm{calc}}$ and $R^{\mathrm{meas}}$ resulted in a mean AAI difference close to zero ($\Delta$AAI = 0.16 while $\sigma = 1.2$), showing that, on average, cloud shadow effects are approximately cancelled out in our

simulated data set.



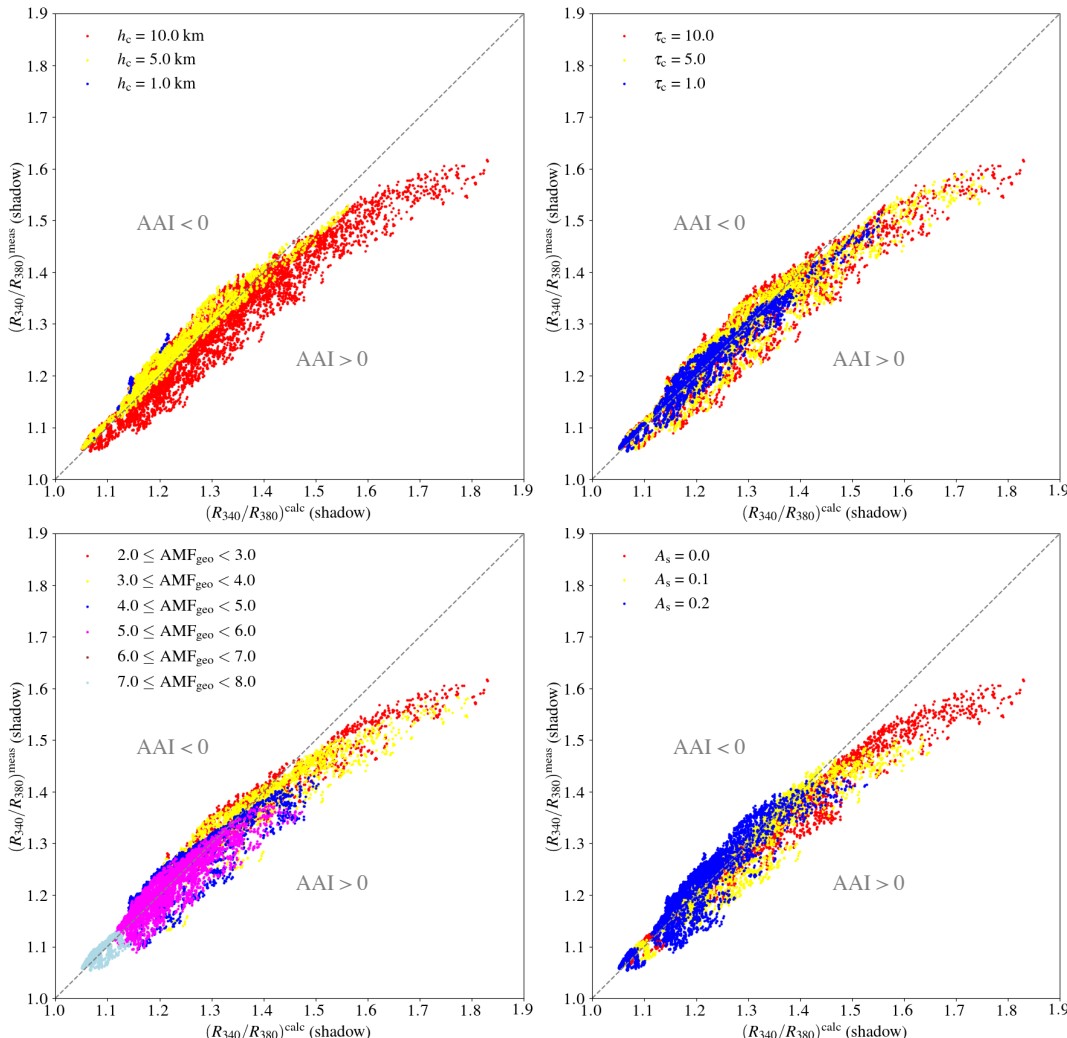

**Figure 9.** Analysis of the simulated measured and calculated TOA reflectance ratios between 340 and 380 nm for the 1350 simulated scenes with MONKI. The results are grouped per cloud height (upper left), cloud optical thickness (upper right), geometric air mass factor (lower left), and surface albedo (lower right).

### 3.3.2 Second order cloud shadow effects

Although the simulated average AAI difference between pixels inside and outside the cloud shadow is close to 0, in some cases the cloud shadows leave signatures in the simulated AAI. This can be concluded from the negative and positive tails of the $\Delta$AAI-histogram in Figure 8, indicating that the cancellation was not always perfect and could have values of several AAI

points. Those imperfect cancellations, while both $R^{\mathrm{calc}}$ and $R^{\mathrm{meas}}$ are both more blue in the cloud shadow, were also found in the observations (Sect. 3.2.2). They are the second order cloud shadow effects.





In the TROPOMI observations (Sect. 3.2.2) we found a slight dependency of the second order cloud shadow effects on geometric air mass factor and surface albedo (through the scene albedo of the first cloud- and shadow-free neighbour). We did not analyze the dependency on cloud parameters in the observations, because our cloud shadow detection algorithm DARCLOS does not allow for a precise determination of the clouds responsible for certain cloud shadows. Figure 9 shows $R_{340}^{\mathrm{meas}}/R_{380}^{\mathrm{meas}}$ and $R_{340}^{\mathrm{calc}}/R_{380}^{\mathrm{calc}}$ for all shadow pixels in our simulated scenes, grouped per cloud height (upper left), cloud optical thickness (upper right), geometric air mass factor (lower left) and surface albedo (lower right). Data points on the symmetry line $R_{340}^{\mathrm{meas}}/R_{380}^{\mathrm{meas}} = R_{340}^{\mathrm{calc}}/R_{380}^{\mathrm{calc}}$ are related to an AAI of 0 (see Eq. (2)) just as in the cloud- and shadow-free region in our simulations, and to a perfect cancellation of cloud shadow effects. Data points in the regions for which $R_{340}^{\mathrm{meas}}/R_{380}^{\mathrm{meas}} > R_{340}^{\mathrm{calc}}/R_{380}^{\mathrm{calc}}$ and $R_{340}^{\mathrm{meas}}/R_{380}^{\mathrm{meas}} < R_{340}^{\mathrm{calc}}/R_{380}^{\mathrm{calc}}$ result in cloud shadow signatures with negative and positive AAI, respectively. Regardless of the grouping of the data points, Fig. 9 demonstrates that the strongest positive AAI cloud shadow signatures are caused by the bluest shadows: when $R_{340}/R_{380} \gtrsim 1.4$ the data points deviate towards the right from the symmetry line. The negative AAI signatures seem to be caused by cloud shadows with $R_{340}/R_{380} \lesssim 1.4$.

Figure 9 (upper left) shows that the strong positive AAI cloud shadow signatures are primarily caused by high clouds ($h_{\mathrm{c}}$ = 10 km). Note that the low clouds ($h_{\mathrm{c}}$ = 1 km) group only contain few data points, because for too small viewing- and or illumination geometries the size of the cloud shadow is too small to be visible from space. Figure 9 (upper right) shows that the thick clouds ($\tau_{\mathrm{c}}$ = 10) give stronger (negative and positive) AAI cloud shadow signatures than thin clouds ($\tau_{\mathrm{c}}$ = 1). Figure 9 (lower left) shows that the AAI cloud shadow signatures tend to become more positive with decreasing geometric air mass factor, which was also found in the TROPOMI observations (see Fig. 6). Figure 9 (lower right) shows that the AAI cloud shadow signatures tend to become more negative with increasing surface albedo, which is also consistent with the TROPOMI observations.

The explanation of the positive second order cloud shadow AAI signature can be found by analyzing the differences in the shadow simulation results between a low and high cloud. Figure 10 shows a top view of the AAI (first row), vertical profiles of the simulated altitude dependent $R_{340}^{\mathrm{meas}}/R_{380}^{\mathrm{meas}}$ contribution (second row), cross-sections of $R_{340}^{\mathrm{meas}}/R_{380}^{\mathrm{meas}}$ and $R_{340}^{\mathrm{calc}}/R_{380}^{\mathrm{calc}}$ at TOA (third row), and the vertical profiles of the contributions of $R_{340}^{\mathrm{meas}}$ and $R_{380}^{\mathrm{meas}}$ at several locations for $x$ (fourth row). The left column shows the results again for a cloud at 5 km altitude (similar as in Fig. 7), while in the middle column the cloud is raised to an altitude of 10 km. For this higher cloud, the AAI is increased up to $\sim$3 points in the shadow, but only close to the cloud ($x$ = 74 km) where the shadow is located in the atmosphere above $\sim$3 km altitude. Here, more light is being intercepted than at lower altitudes, since the contribution to the TOA signal in the shadow-free background ($x$ = 26 km) at these higher altitudes is larger than close to the surface (see orange lines in the bottom figures in Fig. 10). Hence, $A_{\mathrm{scene}}$ is much darker than for the low cloud at $x$ = 74 km, which strongly increases $R_{340}^{\mathrm{calc}}/R_{380}^{\mathrm{calc}}$ (see Sect. 2.2). Also, $R_{340}^{\mathrm{meas}}/R_{380}^{\mathrm{meas}}$ is larger for this shadow at higher altitudes, as multiple scattering contributions at those higher altitudes are more blue (see fourth row in Fig. 7) and the shadow only contains multiply scattered light (see Sect. 3.3.1). However, the latter effect on the vertically integrated signal is relatively weak, because the single scattering background contribution that is being intercepted is also more blue at higher altitudes than at lower altitudes (cf. third row in Fig. 7), which suppresses the increase in measured blueness of the high shadow as seen from space. The suppression of the increase in measured shadow blueness is most effective at small geometric

**Figure 10.** Simulations by MONKI of the AAI (first row), vertical profiles of the simulated altitude dependent $R_{340}^{\mathrm{meas}}/R_{380}^{\mathrm{meas}}$ TOA contribution (second row), cross-sections of $R_{340}^{\mathrm{meas}}/R_{380}^{\mathrm{meas}}$ and $R_{340}^{\mathrm{calc}}/R_{380}^{\mathrm{calc}}$ at TOA (third row), and the vertical profiles of the TOA contributions of $R_{340}^{\mathrm{meas}}$ and $R_{380}^{\mathrm{meas}}$ (fourth row) at $x = 26$ km (unshadowed region), $x = 74$ km (atmosphere shadow), and $x = 94$ km (surface shadow). The scene parameters in the first column are similar as in Fig. 7: $h = 5$ km, $\tau_{\mathrm{c}} = 10$, $r_{\mathrm{eff}} = 2.0$ $\mu$m, $A_{\mathrm{s}} = 0.0$, $\theta_0 = 75°$, $\theta = 0°$, $\varphi - \varphi_0 = 0°$. In the second row, $h$ was modified into 10 km. In the third row, in addition $A_{\mathrm{s}}$ was modified into 0.2. The vertical profiles were made using the mean values from $y = 30$ to $y = 46$ km. The data point below 0 km altitude represents the contribution of the surface.





AMF, for which the vertical profiles of the single scattering background contribution are more blue due to the relatively short path lengths (not shown), and because the contribution peaks at lower altitudes (i.e., the lower atmosphere, where the shadows usually occur, are better visible from space). Consequently, the positive second order cloud shadow signature increases with

decreasing geometric AMF. In summary, both $R_{340}^{\mathrm{meas}}/R_{380}^{\mathrm{meas}}$ and $R_{340}^{\mathrm{calc}}/R_{380}^{\mathrm{calc}}$ at TOA are larger for shadows located higher in the atmosphere, but $R_{340}^{\mathrm{calc}}/R_{380}^{\mathrm{calc}}$ increases stronger with altitude than $R_{340}^{\mathrm{meas}}/R_{380}^{\mathrm{meas}}$, causing the AAI to become positive according to Eq. 2.

The explanation of the negative second order cloud shadow AAI signature can be found by increasing the surface albedo $A_{\mathrm{s}}$ from 0 to 0.2 (see third column in Fig. 10, and Fig. A1 for more detailed simulation results). Without shadows, wave-

length independent Lambertian surface reflection makes the signal at TOA stronger but more 'white', resulting in both smaller $R_{340}^{\mathrm{meas}}/R_{380}^{\mathrm{meas}}$ and $R_{340}^{\mathrm{calc}}/R_{380}^{\mathrm{calc}}$ (see upper and lower right figures, respectively, in Fig. A1), and a neutral effect on the AAI (see de Graaf et al., 2005) (see lower right figure in Fig. A1). In our simulations, at a relatively large distance from the cloud where the shadow is cast on the surface (at $x = 94$ km), the AAI is decreased by $\sim 1.5$ points. Here, incident light on the surface is being reflected, resulting in a larger TOA reflectance compared to that in the black surface case. Consequently, $A_{\mathrm{scene}}$ in

the AAI retrieval is higher (see bottom middle figure in Fig. A1), such that the Lambertian surface in the DAK model reflects more direct and scattered scattered light, which relatively decreases $R_{340}^{\mathrm{calc}}/R_{380}^{\mathrm{calc}}$. However, in reality in the cloud shadow on the surface at $x = 94$ km, the surface only reflects light that has been scattered before at least once (see third row in Fig. A1). Because the multiply scattered light contribution is more blue (see fourth row in Fig. A1) than that of singly reflected light by the surface in the background, the measured surface shadow is relatively blue, which increases $R_{340}^{\mathrm{meas}}/R_{380}^{\mathrm{meas}}$. In summary,

the Lambertian surface reflection decreases both $R_{340}^{\mathrm{meas}}/R_{380}^{\mathrm{meas}}$ and $R_{340}^{\mathrm{calc}}/R_{380}^{\mathrm{calc}}$ in the cloud shadow cast on the surface, but because the decrease of $R_{340}^{\mathrm{calc}}/R_{380}^{\mathrm{calc}}$ is stronger the AAI becomes negative.

## 4   Discussion and conclusion

The cancellation of cloud shadow effects on both the measured and simulated TROPOMI AAI (Sect. 3.2 and 3.3, respectively) shows that the traditional AAI retrieval by itself already (partly) corrects for cloud shadows via the retrieved scene albedo.

Simultaneously, we measure and simulate that cloud shadows are in principle always more blue than cloud- and shadow-free regions. If the traditional AAI retrieval would not correct for this enhanced blueness, strong cloud shadow signatures could have been expected in the AAI. But, due to the automatic cancellation, the average AAI difference between shadow and non-shadow cases is close to zero.

We have shown that, for individual cases in the measurements and in the simulations, the blueness of the clouds shadows

is not always perfectly compensated for by the lower scene albedo in the AAI retrieval. This results in second-order cloud shadow effects which sometimes yield lower, and sometimes higher, AAI than in the cloud- and shadow-free regions. In the observations, we found weak positive and negative relations of those second order cloud shadow effects to the geometric air mass factor and surface albedo, respectively. Our simulations indeed demonstrated that positive AAI cloud shadow signatures





can mostly be related to high thick clouds with small geometric air mass factor above dark surfaces, while negative AAI cloud
shadow signatures should be most prominent near thick clouds above bright surfaces.

Our simulations thus suggest that a potential correction of the second order cloud shadow effects on the AAI should depend
on cloud height, optical thickness, surface albedo and geometric air mass factor. However, the height and thickness of the clouds
responsible for the measured cloud shadows are uncertain. That is because, although the cloud height is a TROPOMI product
(e.g. FRESCO, see Koelemeijer et al., 2001; Wang et al., 2008), the cloud height product has a limited accuracy (the cloud
height obtained with FRESCO is in fact the cloud centroid height) and the optical thickness and vertical extent of the clouds
are not retrieved. Moreover, the clouds responsible for certain cloud shadows are difficult to determine in the observations. The
responsible clouds are not an output of DARCLOS, as DARCLOS uses spectral tests to determine the cloud shadow flags in the
final step of its algorithm. Additionally, the accuracy of a 'reverse calculation' of the responsible cloud (height) would never be
better than the ∼4 km spatial resolution of TROPOMI in the nadir viewing direction, and again the cloud optical thickness and
vertical extent would be unknown. Hence, we conclude that a reliable correction method for the second order cloud shadow
effects on the TROPOMI AAI would be complicated. Moreover, because of the automatic cancellation of the cloud shadow
effects to the first order, such a correction method may not be needed.

For this study, we have developed the 3D radiative transfer code MONKI which successfully simulated the effect of cloud
shadows on the TROPOMI AAI. MONKI fully takes into account the polarization of light for all orders of scattering, and can
store the vertical profiles of the altitude dependent reflected light contribution at TOA, for the total, singly, and multiply scat-
tered light. In future research, MONKI can be used to find explanations of more cloud effects on sensitive retrieval algorithms,
such as the AAI algorithm, in which polarization and geometry play on important role. For example, the positive AAI increases
at the bright side of clouds that are found in both our simulations and previous observations, can be further analyzed using the
MONKI model.

**Appendix A: MONKI simulations of a shadow cast on a reflecting surface produced by a high cloud**

Figure A1 shows the MONKI output results as in Fig. 7, but then for a high cloud ($h = 10$ km) above a reflecting Lambertian
surface ($A_\mathrm{s} = 0.2$). Although cloud shadows in the atmosphere and on the surface are more blue than their shadow-free sur-
roundings in both $R_{340}^\mathrm{meas}/R_{380}^\mathrm{meas}$ and $R_{340}^\mathrm{calc}/R_{380}^\mathrm{calc}$ (first order effect), the blueness is not increased equally resulting in a positive
AAI signature of $\sim 2.5$ in the atmosphere close to the cloud, at $x = 74$ km, and a negative AAI signature of $\sim -1.5$ in the
shadow cast on the surface, at $x = 94$ km (second order effects).

*Author contributions.* V.T. did all computations and wrote the manuscript. P.W. weekly commented on the intermediate results. All authors
read the manuscript, provided feedback that led to improvements and were involved in the selection of the results presented in this paper.



**Figure A1.** Similar as Fig. 7, but for $h = 10$ km and $A_s = 0.2$.

*Competing interests.* The authors declare that they have no conflict of interest.



*Acknowledgements.* This work is part of the research programme User Support Programme Space Research (GO) with project number
ALWGO.2018.016, which is (partly) financed by the Dutch Research Council (NWO).



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
