# Peer review of "Cancellation of cloud shadow effects in the absorbing aerosol index retrieval algorithm of TROPOMI"

_Atmospheric Measurement Techniques, 2024_

## Referee Comment (RC2)

Review of manuscript 'Cancellation of cloud shadow effects in the absorbing aerosol index retrieval algorithm of TROPOMI' by Trees et al submitted for publication in AMT

**Summary**

In this paper the authors discuss the effect of cloud shadows in the magnitude of the qualitative Absorbing Aerosol Index (AAI) parameter derived from TROPOMI near UV observations and verify their findings with 3D radiative transfer (RT) calculations. The reported RT results are a good validation of their developed 3-D RT tool application to UV observations. They conclude that the effect on the AAI is quite small and that, therefore, a correction is not necessary.

**General Comments**

The detection of cloud shadows signal in the AAI is not a new finding. Cloud shadows in equivalent AAI definitions have been previously identified in satellite observations by sensors with UV observational capability at sub-kilometer spatial resolution [Fukuda et al., 2013; Tanada et al., 2023; Gogoi et al, 2023]. As in the previously published analyses, this manuscript concludes that the cloud shadow effect on the AAI is quite small and does not warrant a correction. This conclusion can be further justified by the fact that the AAI is not a physically meaningful parameter. Thus, in its current form, the manuscript's main contribution is the validation of the MONKI UV radiative transfer scheme and could, perhaps, be submitted to an RT specialized journal.

The authors could advantageously use the documented AAI-cloud shadow detection capability to develop corrections to radiance measurements at relevant channels in the UV-VIS-near-IR range that are used in the retrieval of important geophysical parameters. Although corrections would not be necessary for DOAS (Differential Optical Absorption Spectroscopy) based applications, they would be needed for cloud and aerosol algorithms and other inversion schemes based on the interpretation of discrete channel observations. One such AAI-based reflectance correction was developed and applied to GOSAT/TANSO CAI observations at 672 nm (Fukuda et al., 2013). Similar applications to TROPOMI measured radiances would be an important contribution to improve the accuracy of retrieved physical parameters and contribute towards the error budget analyses of TROPOMI retrieved parameters. I encourage the authors to pursue this goal.

**Specific Comments**

Line 19: Suggest replacing *spectrometer* with *hyperspectral instrument.*

Line 22: Suggest adding *hyperspectral* before *predecessors*.

Line 24: It can be said that TROPOMI's 3.6X5.6 km spatial resolution is indeed unprecedented in hyperspectral sensors. However, that is not the case for multi-wavelength sensors. Near UV channels at sub-kilometer resolution have been added to several sensors over the last decade. In 2009, the Cloud Aerosol Imager (CAI) sensor on the Japanese GOSAT TANSO satellite made 380 nm radiance measurement at 0.25 km resolution (Fukuda et al., 2013). Similar measurements were

carried out by the Second-generation Global Imager (SGLI) launched in 2017 (Tanada et al, 2023). Measurements at 339 and 377 nm at 0.46 km resolution have been made by CAI-2 since its launch in 2018 (Gogoi et al., 2023). The recently deployed PACE-OCI instrument measures UV radiances at 1 km spatial resolution (Werdell et al., 2019).

Lines 34-35: Another important non-aerosol related effect on measured near UV radiances is pure water absorption over the remote oceans. Suggest adding '*as well as pure water absorption effects over the remote oceans (Fry, 2000)*' after (Kooreman et al., 2020).

Line 39: Suggest adding *hyperspectral* before *predecessors*.

Line 40: A brief discussion on earlier studies on cloud shadows in the near UV applied to observations by other high spatial resolution multi-wavelength instruments (JAXA's CAI, SGLI and CAI-2) should be included here.

Line 99. Suggest qualifying or removing the statement '*In the absence of aerosols and clouds, the AAI is, in theory, ideally equal to zero'*. Actually, there are multiple non-aerosol related effects such as land surface reflectance spectral effects (deserts in particular) and ocean signal associated with sunglint and clear water absorption (both yielding positive AAI values) as well as chlorophyll absorption that yields negative AAI values. An explanation of how these non-aerosol related effects are detected and flagged should be included.

Line 100 Elaborate on the reasons for such a large negative offset. Is the reported AAI adjusted for this offset? If so, how? Provide references on this issue.

Line 101 On the detectability of aerosols above clouds suggest adding the Torres et al (2012) reference.

**References**

Fry, E.S., Visible and near ultraviolet absorption spectrum of liquid water. Appl. Opt. 2000, 39, 2743–274

Fukuda, S., T. Nakajima, H. Takenaka, A. Higurashi, N. Kikuchi, T. Y. Nakajima, and H. Ishida (2013), New approaches to removing cloud shadows and evaluating the 380nm surface reflectance for improved aerosol optical thickness retrievals from the GOSAT/TANSO-Cloud and Aerosol Imager, J. Geophys. Res. Atmos., 118, 13,520–13,531, doi:10.1002/2013JD020090.

Gogoi, M. M., Babu, S. S., Imasu, R., and Hashimoto, M.: Satellite (GOSAT-2 CAI-2) retrieval and surface (ARFINET) observations of aerosol black carbon over India, Atmos. Chem. Phys., 23, 8059–8079, https://doi.org/10.5194/acp-23-8059-2023, 2023.

Tanada, K., Murakami, H., Hayasaka, T., & Yoshida, M. (2023). Aerosol optical properties of extreme global wildfires and estimated radiative forcing with GCOM-C SGLI. *Journal of Geophysical Research: Atmospheres*, 128, e2022JD037914.
 https://doi.org/10.1029/2022JD037914

Torres, O, H. Jethva, and P.K. Bhartia, Retrieval of Aerosol Optical Depth above Clouds from OMI Sensitivity Analysis and Case Studies, Journal. Atm. Sci., 69, 1037-1053 Observations: Sensitivity Analysis and Case Studies, Journal. Atm. Sci., 69, 1037-1053, doi:10.1175/JAS-D-11-0130.1, 2012

Werdell, P.J., Behrenfeld, M.J., Bontempi, P.S., Boss, E., Cairns, B., Davis, G.T., Franz, B.A., Gliese, U.B., Gorman, E.T., Hasekamp, O., Knobelspiesse, K.D., Mannino, A., Martins, J.V., McClain, C.R., Meister, G., and Remer, L.A. (2019). The Plankton, Aerosol, Cloud, Ocean Ecosystem Mission: Status, Science, Advances, Bull. Am. Meteorol. Soc., 100(9), doi: 10.1175/BAMS-D-18-0056.

---

## Author Response (AR1)

**Response to comment of Referee #1 on "Cancellation of cloud shadow effects in the absorbing aerosol index retrieval algorithm of TROPOMI" by Victor Trees et al.**

Victor J. H. Trees [1,2], Ping Wang [1], Piet Stammes [1], Lieuwe G. Tilstra [1], David P. Donovan [1,2], and A. Pier Siebesma [2]

[1]Research & Development Satellite Observations, Royal Netherlands Meteorological Institute (KNMI), Utrechtseweg 297, 3731 GA, De Bilt, the Netherlands
[2]Department of Geoscience & Remote Sensing, Delft University of Technology, Stevinweg 1, 2628 CN, Delft, the Netherlands

**Correspondence:** Victor Trees (victor.trees@knmi.nl)

We thank the reviewer for his/her careful reading and for the comments and suggestions, which have improved the manuscript. Below, we give in *blue italic* the reviewer's comment, in black our response, in *black italic* copied text from the manuscript and in *red italic* the changed or new text in the manuscript.

*The manuscript describes the findings of the systematic investigation of the effect that cloud shadows have on the AAI. The investigation was performed on an observational data set and compared with results from radiative transfer model (RTM) calculations using a newly developed Monte Carlo model. The authors conclude that cloud shadow effects on AAI are (1) difficult to correct, and (2) can mostly be ignored (due to the cancellation occurring within the AAI definition). The manuscript addresses a topic that could be neglected when the AAI was originally developed, but has become of importance with the*

*advent of TROPOMI, which has a footprint on the same order of magnitude as cloud effects.*

*The investigation is well performed, the manuscript reads well and figures are illustrative and appropriate. Nevertheless, I have a number of serious concerns that I believe should be addressed before the manuscript can be published.*

*First: the advantages of using a Monte Carlo RTM to aid the understanding of cloud shadows and their effects on AAI can be clearly seen in the Figures 7 and 10, as the atmosphere's extinction profile can be studied. However, the model is not described in any detail (not even the acronym is explained) and the reader is expected to rely solely on the authors' statement that the RTM "shows an excellent agreement" with DAK (line 177). To lend credibility to the results shown in the manuscript, at least part of the analysis should be performed with DAK and the results compared.*

We thank the reviewer for the request to elaborate on our Monte Carlo code MONKI. The acronym was explained on line 164:

    *"... In this research, we use the three-dimensional radiative transfer code MONKI (Monte Carlo KNMI)."*

Unfortunately, it is not possible to redo the analysis with DAK, because DAK is a radiative transfer code assuming plane parallel atmospheric layers (i.e., without 3D-ness), and therefore cannot simulate cloud shadows. Section 2.6 provides a short explanation of MONKI, but does indeed not explain the model in detail, in order to avoid a loss of focus. We are currently preparing a publication about MONKI, which contains comparison results between MONKI and DAK (for 1D scenes) and MONKI and other 3D Monte Carlo codes using the code intercomparison paper of Emde et al. (2018).

The good agreement between MONKI and DAK is also illustrated in Fig. 7. Outside the cloud and the shadow, the AAI is virtually equal to zero ($-0.01$ on average with a standard deviation of $0.07$ due to Monte Carlo noise). This means that in those regions the simulated measured TOA reflectances at 340 and 380 nm with MONKI, $R_{340}^{\mathrm{meas}}/R_{380}^{\mathrm{meas}}$, are close to the calculated TOA reflectances with DAK, $R_{340}^{\mathrm{calc}}/R_{380}^{\mathrm{calc}}$, in Eq. 2.

*Second: a number of RTM results are presented and explained, but the explanations are not always easy to follow. A schematic diagram of the phenomena involved, combined with a conceptual explanation, would greatly improve the understanding. The diagram could show, in a more simple way than Figs. 7 and 10, what the effects of surface, atmosphere, and clouds are on the reflectance and how this results in the observed (and modelled) blueing effect. This would particularly aid the understanding of "second-order effects", which I had trouble with.*

We have added a conceptual model to the paper, explaining the cancellation of cloud shadow effects on the AAI, as shown in Fig. 1 of this reply. We did not add diagrams for the second order cloud shadow effects, because that would become very messy with all the arrows involved. We agree that the explanation of the second order cloud shadow effects is long. The second order cloud shadow effect can unfortunately not be explained with a few sentences, or be captured by a simple diagram. We think Fig. 7 and 10 in their current forms are the best option to support the text.

*Third: the discussion of the manuscript is very limited. The authors argue at length why a shadow-effect correction to the AAI is not feasible (or necessary), a minor aspect of the manuscript, whilst ignoring a number of important aspects. The following points need to be added:*

*(1) statistics as to how often serious AAI deviations (greater than, say, 1 unit) due to high thick clouds, high surface albedo, and/or low AMF_geo are encountered in observations. This would provide a more objective basis on which to build the argument that the correction is or is not necessary.*

We thank the reviewer for this suggestion. The cloud shadow detection algorithm DARCLOS does not identify the clouds that were responsible for the cloud shadows, so the corresponding cloud optical thickness and cloud height are uncertain in the observations (the cloud height was used in DARCLOS to compute the potential cloud shadow flag, but it was not used in the refinement to the spectral cloud shadow flag). However, in the paper we studied the dependency of the AAI cloud shadow effect on the geometric air mass factor and on the scene albedo. We have added the following lines:

[Figure]

**Figure 1.** Sketches explaining the first order cloud shadow effect on the Absorbing Aerosol Index (AAI). The top and bottom sketches are for the measured (meas) and the calculated (calc) top-of-atmosphere (TOA) reflectances, respectively. The left four sketches are for the clear case (i.e., without clouds and shadows), and the right four sketches are for the cloud shadow case, where the first row is for 340 nm and the second row for 380 nm. Solid arrows indicate singly scattered light, dashed arrows indicate multiply scattered light, and dotted arrows indicate light reflected once by the surface. The number of arrows leaving TOA illustrate the magnitude of the respective TOA reflectance. The AAI retrieval algorithm automatically assumes a dark surface when the measured reflectance is low due to the shadow (Eq. 4), as illustrated by the black shaded surface area. Because cloud shadows increase the ratio of the TOA reflectance at 340 nm with respect to 380 nm in both the measurements and the retrieval algorithm calculations by approximately the same amount, the AAI is more or less unaffected (Eq. 2).

Paragraph starting on line 248:

*We count an increase of 377 pixels for which $\Delta AAI > 1$, when cloud shadow pixels instead of second neighbour pixels are compared with first neighbour pixels, provided that $AMF_{geo} < 5$. This number corresponds to $0.47\%$ of the total number of shadow pixels.*

*We count an increase of 70 pixels for which $\Delta AAI < -1$, when cloud shadow pixels instead of second neighbour pixels are compared with first neighbour pixels, provided that $A_{\text{scene}} > 0.2$. This number corresponds to $0.09\%$ of the total number of shadow pixels.*

Also, we have added to the conclusion, on line 405:

*In the observations, $0.47\%$ and $0.09\%$ of the shadow pixels show an absolute AAI difference larger than 1, with respect to their cloud- and shadow-free neighbours, that can be attributed to the cloud shadow, when selection data with $AMF_{geo} < 5$*

*and $A_{\text{scene}} > 0.2$, respectively.*

*(2) A discussion of the effects of aerosols on the studied shadow effects — AAI is an aerosol index, after all — and*

The focus of the paper is on the features that cloud shadows leave in the AAI, regardless whether or not there are aerosols in the scene. In that extent it is an extension of the paper by Kooreman et al. (2020), who focused on the effect of clouds on the AAI, regardless of the presence of absorbing aerosols. We think that a complete analysis of the behaviour of the AAI in scenes with both absorbing aerosols and clouds (and their shadows) would fit better in another publication. Therefore, we consider it as beyond the scope of this paper. We note that only 8 shadow pixels (0.01 % of the shadow pixels) may also contain absorbing aerosol (based on a AAI $> 0.8$ threshold for the cloud- and shadow-free neighbour pixels, see de Graaf, 2022). We have added the sentence to the conclusion to emphasize this limitation of our work:

*We did not specifically select scenes that also include absorbing aerosol for this paper. We note that only 0.01% of the shadow*

*pixels also may contain absorbing aerosols (based on a AAI $> 0.8$ threshold for the cloud- and shadow-free neighbour pixels, see de Graaf, 2022).*

*(3) the opposite: the effect of blueing on TROPOMI aerosol retrievals. Even if a detailed discussion is out of scope, the issue should be mentioned in your discussion.*

We have added the following sentence to line 398:

*We note that other TROPOMI products that depend on the pixel blueness, such as the aerosol optical thickness (AOT) (de Graaf, 2022), may be affected by cloud shadows, but that was not studied in this paper.*

*(4) A discussion on how the choice of first and second neighbours influences the observational analysis: E.g., Fig. 10 shows*

*that pixels between the cloud and the shadow have a larger AAI deviation than those in the shadow. And what happens if neighbouring pixels with a strongly deviating surface albedo are selected for comparison?*

The positive AAI values in between the surface shadow and the cloud in Fig. 10 belong to the shadow inside the atmosphere. The shadow inside the atmosphere is also darker than the cloud- and shadow-free surrounding (see e.g. the $A_{\text{scene}}$ values in Fig. 7), and should therefore be flagged as a shadow pixel when $\Gamma(\lambda = 380)$ nm $< -15\%$. If atmosphere shadows are not dark enough to reach this darkness threshold, we at least make sure that they are not flagged as shadow-free neighbour pixel, by deleting the potential neighbour pixels from the search area if (see Sect. 2.5):

$$\Gamma(\lambda = 380) \text{ nm} < 0\% \tag{1}$$

Hence, we do not expect the atmosphere shadows to affect our results, as they are either flagged as shadow pixels or deleted from search areas for neighbour pixels.

The first and second neighbour pixel indeed may contain scenes that are not comparable to those of the shadow pixels. Therefore, we also have presented the AAI comparison between the first and second neighbour pixels in e.g. the first column of

Fig. 4. Those scatter diagrams represent the natural variation that cannot be attributed to cloud shadow, such as surface albedo differences. If a first neighbour pixel with a strongly deviating surface albedo is selected, it could increase the AAI difference between the shadow and the first neighbour pixel. But, we believe there is an equal probability that the shadow pixel itself, or the second neighbour pixel, encounters a strongly deviating surface albedo. Therefore, we think that our approach to estimate the cloud shadow impact is unbiased.

*All in all, the manuscript contains the description of a number of interesting observations and calculations, but in its present*
*form does not advance the field. A more in-depth analysis of the results, coupled with sufficient evidence that MONKI is an appropriate tool for the study, would greatly improve the impact of the research.*

*Please find a few other comments and suggestions below:*

*l. 69: "extraterrestrial solar irradiance perpendicular to the beam" - change to "solar irradiance"*
We have removed 'extraterrestrial' and 'perpendicular to the beam', as suggested.

*ll. 109-110: "As shown in Figure 1, (...) overlap each other." - change to "The area of interest in Europe was covered by TROPOMI during three successive, partially overlapping overpasses on November 11, 2020, as shown in Fig. 1."*
We thank the reviewer for the suggestion, which indeed reads better. We have replaced the sentence by the suggested sentence.

*ll. 110-112: "For each day (...) from the data set." - insert the sentence before "The selected" on line 106*
We have moved the sentence to line 106, as suggested.

*ll. 116-117: "the already available effective cloud fraction in the TROPOMI NO2 product" - Which cloud algorithm does that come from - FRESCO?*
This effective cloud fraction does not come from FRESCO, but from the cloud retrieval of the $NO_2$ processor. This is because the effective cloud fraction depends on wavelength and $NO_2$ is retrieved at shorter wavelengths than normally considered in FRESCO. The reference to the $NO_2$ ATBD (van Geffen et al., 2021) is already provided in the same sentence, in which the
reader can find the details of this implementation.

*l. 130 - "SCSFs are a better estimate of the cloud shadows than the PCSFs." - In which sense are they better: more accurate? Less or more strict? Where is the evidence? The next sentence refers the reader to the right panel of Fig. 1 "as an example", but it is not clear what one should see there.*
We agree with the reviewer that this part was not clearly formulated. We have added a figure of the PCSF as Fig. 2.b to Fig. 2 as follows:

[Figure]

**Figure 2.** The scene albedo at 380 nm, $A_{\text{scene}}(\lambda = 380 \text{ nm})$, derived by TROPOMI on 3 November 2020 above the Netherlands, Belgium and North-West Germany (Fig. 2a), the potential cloud shadow flags (PCSFs) in blue, spectral cloud shadow flags (SCSFs) in red, cloud flags (CFs) in white (Fig. 2b), and the first cloud- and shadow-free neighbour pixels in yellow and possibly shadow affected pixels according to Eq. (7) in blue (Fig. 2c).

Comparing Figs. 2.a and 2.b shows that the PCSF often overestimates the cloud shadow area, while for every SCSF we find a dark pixel in $A_{\text{scene}}$. We have changed the text as follows:

Line 130: *... shows the SCSFs indicated in red, the PCSFs indicated in blue and the CFs indicated in white, in three TROPOMI orbits covering the area of our case study, at 3 November 2020 which is one of the days in our data set. Figure 2b shows the SCSFs and PCSFs zoomed in on North-West Germany. From visual comparison of Fig. 2b to the map of the scene albedo $A_{\text{scene}}$ (Fig. 2a), it may be observed that the SCSFs are indeed located at pixels where $A_{scene}$ is lower than at surrounding pixels along cloud edges, which may be interpreted as cloud shadows. For more details about the cloud shadow flagging with*

*DARCLOS, we refer to Trees et al. (2022).*

*l. 141: " potential neighbour pixels of two TROPOMI pixels" - change to "potential neighbour pixels within a two-pixel radius".*

We thank the reviewer for the suggestion. We have changed the sentence as follows:

l. 141:  –> *First, for each shadow pixel, we define a search area with potential neighbour pixels within a two-pixel radius around the shadow pixel.*

*l. 143: " Because we require (...) shadow free," - redundant, can be removed*

We have removed this half sentence, as suggested.

*l. 198: " (2) increased shadow darkness due to the longer slant path length of the incoming direct light through the clouds."*
*- Is this hypothesis based on calculations? Intuitively, I'd say that there is a larger amount of indirect (Rayleigh-scattered)*
*radiation as well, maybe counter-acting this effect. Also, the light-path length effect is only valid for relatively thin clouds, as*
*thick clouds will not let any radiation through anyhow.*

We agree with the reviewer that multiple scattering may counteract the shadow darkness at large solar zenith angles. We thank the reviewer for pointing this out and we have removed this sentence.

*Fig. 4, upper right panel: please shrink the x-axis to -/+ 2.5 units to make out more details.*

We have changed the figure as suggested.

*Fig. 4, middle and lower right panels: is the secondary peak visible at Delta(R340/R380)= 0 real? Where does it come from?*

There indeed seems to be a secondary peak in the histograms of $\Delta\left(R_{340}/R_{380}\right)^{\mathrm{meas}}$ and $\Delta\left(R_{340}/R_{380}\right)^{\mathrm{calc}}$, centered at about 0. We have analyzed the values of $\Delta\left(R_{340}/R_{380}\right)^{\mathrm{calc}}$ (for which the secondary peak seems strongest), as functions of viewing and illumination geometry. We found that the values of $\Delta\left(R_{340}/R_{380}\right)^{\mathrm{calc}}$ close to zero are found at very negative viewing zenith angles $< -55$ degrees, as shown in Fig. 3 of this reply. Due to the longer path lengths, those scenes are relatively white ($(R_{340}/R_{380})^{\mathrm{calc}}$ is relatively low) regardless of the shadows, and differences in $(R_{340}/R_{380})^{\mathrm{calc}}$ between the shadow and surroundings are relatively small. We found that those points occur throughout the whole study area. We could not find another explanation for why there seems to be a high concentration of shadow pixels at those negative viewing zenith angle other than coincidence.

[Figure]

**Figure 3.** The $(R_{340}/R_{380})^{\mathrm{calc}}$ differences between the cloud shadow pixels and first neighbour pixels (left) and between the second and first neighbour pixels (right), as functions of viewing zenith angle. The black arrow points at the concentration of points with $(R_{340}/R_{380})^{\mathrm{calc}}$ close to zero and very negative viewing zenith angles ($< -55$ degrees).

*Section 3.2.2: This section is hard to follow without a conceptual diagram, as suggested above*

As discussed above, we have added a conceptual diagram for the first order cloud shadow effect. We agree that the explanation of the cause of the second-order cloud shadow effects is long, but we think it could unfortunately not be replaced by a simple diagram or a short explanation, as that would not cover the full explanation.

*l. 273 and further: "gas" - change to "atmosphere"*

The Rayleigh scattering optical thickness only applies for the gas. Therefore, we keep the word 'gas' instead of the word 'atmosphere' in this sentence, which emphasizes that the cloud droplets are not Rayleigh scatterers at our wavelengths of interest.

 *l. 278: "gas pressure (...) is largest." - change to: "atmosphere is most dense."*

We have changed the sentence as follows:

l. 278:  –> *where the gaseous atmosphere is most dense.*

 *l. 280: "of the gas" - remove*

We keep 'of the gas' in this sentence, because in the previous sentence we removed 'Rayleigh scattering optical thickness' and only mentioned 'the gaseous atmosphere'. 'Of the gas' now refers to 'the gaseous atmosphere' in the previous sentence.

*ll. 296-297: " Apparently, all photons were scattered away from the direct beam" - this is not surprising in view of the cloud*
 *droplets' Mie phase function, which features a strong forward-scattering peak*

The Mie phase function describes the scattering direction preference of light upon scattering by the cloud droplets, but it does not describe the chance of scattering away from the direct beam (i.e., the extinction due to scattering). The extinction is determined by the optical thickness of the cloud and the direction of propagation of the direct beam through the cloud. Hence, we do not think that the Mie phase function contains information about the photon survival along the direct beam through the cloud.

*Fig. 7, lower left panel: please change the color scale to a more appropriate range, like -2.5 to 2.5; it would make the effects more apparent to the reader*

We have chosen this color scale to be consistent with 10 AAI points wide range used in Figs. 1 and 5, which is the range commonly used in observations to distinguish absorbing aerosol, see e.g. Kooreman et al. (2020) and https://mpc-l2.tropomi.
 eu/maps.html#aerosol_index_354_388_2. Therefore, we have decided to keep this color scale.

*Fig. 7, lower center panel: the shadow is very difficult to pick out in the figure; it might be helpful to change the color range*

We have changed the plotting range, from [-0.1;0.4] to [-0.1;0.3], which makes the cloud shadow better visible.

 *Fig. 8: what do the histograms show? The number of counts is appreciably higher than the number of scenes.*

The count is indeed higher than the number of scenes, because a scene can have multiple cloud shadow pixels. The caption of the figure already mentions that the histograms represent the differences between the cloud shadow pixels versus the cloudand shadow-free pixels. In order to make it even more clear, we have added the sentence to the caption of Fig. 8:

*The total count is higher than the number of scenes, because a scene can have multiple cloud shadow pixels.*

*Fig. 9: these plots would make more sense as AAI contours plots with h_c and tau_c on the x- and y-axis, resp. Then these plots can disappear into the appendix. It's the AAI you're interested in, not the blueness per se.*

Fig. 9 supports the explanation in the text of the cause of the second order cloud shadow features in the AAI. This explanation is based on the blueness, therefore, Fig. 9 shows the blueness. Additionally, we note that contour plots as functions of $h_c$ and $\tau_c$ would not be so practical for our paper, because only 3 values of $h_c$ and $\tau_c$ are used in our analysis. Therefore, we have decided to keep Fig. 9 as is.

*Section 3.3.2.: This section contains a lot of interesting results that probably form the key to understanding the investigated effects — but I fail to understand it completely. Particularly the paragraph starting on line 378 is difficult to follow without a schematic.*

*ll. 409-410: "the cloud height obtained with FRESCO is in fact the cloud centroid height" - this is a rather lazy argument, as the correction could simply be made to depend on the cloud centroid height.*

The cloud top height determines the vertical cloud size, and with that the maximum shadow extent in the horizontal direction. With only the cloud centroid height, the cloud's vertical size remains unknown. Therefore, we do not think that a correction could simply be made dependent on the cloud centroid height.

*l. 414: "the 4 km spatial resolution" - is this not sufficient for such a correction?*

The ∼4 km spatial resolution is only in the nadir viewing direction of TROPOMI. For slanted observations of the swath, the ground pixel size increases up to 15 km. Because we see significant differences in AAI output in shadows for clouds at 1, 5 and 10 km in our simulations (Fig. 9), we think that the spatial resolution of TROPOMI is not sufficient for a cloud height retrieval based on just the cloud shadow geometry.

*l. 422: "the positive AAI increases": please note that we have observed and modeled the same effect for a high-altitude non-absorbing aerosol plume in the past as well [Penning de Vries et al., 2014]. Reference: Penning de Vries, M. J. M., Dörner, S., Pukīte, J., Hörmann, C., Fromm, M. D., and Wagner, T.: Characterisation of a stratospheric sulfate plume from the Nabro volcano using a combination of passive satellite measurements in nadir and limb geometry, Atmospheric Chemistry and Physics, 14, 8149–8163, https://doi.org/10.5194/acp-14-8149-2014, 2014.*

We think that this AAI increase at the cloud edge could be a 3D-effect caused by the vertical dimension of the cloud and the illumination of the cloud from the side (as explained on lines 317 to 322), rather than a viewing zenith angle dependent effect caused by the cloud droplet's phase function. Such 3D-effects are not discussed in Penning de Vries et al (2014).

**References**

de Graaf, M.: TROPOMI ATBD of the Aerosol Optical Thickness. Doc. No. S5P-KNMI-L2-0033-RP, Issue 3.0.0, Royal Netherlands Meteorological Institute (KNMI), https://data-portal.s5p-pal.com/product-docs/aot/s5p_aot_atbd_v3.0.0_2022-02-10_signed.pdf, [Online; accessed 18-January-2024], 2022.

Emde, C., Barlakas, V., Cornet, C., Evans, F., Wang, Z., Labonotte, L. C., Macke, A., Mayer, B., and Wendisch, M.: IPRT polarized radiative transfer model intercomparison project - Three-dimensional test cases (phase B), Journal of Quantitative Spectroscopy and Radiative Transfer, 209, 19–44, https://doi.org/10.1016/j.jqsrt.2018.01.024, 2018.

Kooreman, M. L., Stammes, P., Trees, V., Sneep, M., Tilstra, L. G., de Graaf, M., Stein Zweers, D. C., Wang, P., Tuinder, O. N. E., and Veefkind, J. P.: Effects of clouds on the UV Absorbing Aerosol Index from TROPOMI, Atmospheric Measurement Techniques, 13, 6407–6426, https://doi.org/10.5194/amt-13-6407-2020, 2020.

Trees, V. J. H., Wang, P., Stammes, P., Tilstra, L. G., Donovan, D. P., and Siebesma, A. P.: DARCLOS: a cloud shadow detection algorithm for TROPOMI, Atmospheric Measurement Techniques, 15, 3121–3140, https://doi.org/10.5194/amt-15-3121-2022, 2022.

van Geffen, J., Eskes, H., Boersma, K., and Veefkind, J.: TROPOMI ATBD of the total and tropospheric $NO_2$ data products. Doc. No. S5P-KNMI-L2-0005-RP, Issue 2.2.0, Royal Netherlands Meteorological Institute (KNMI), https://sentinel.esa.int/documents/247904/2476257/Sentinel-5P-TROPOMI-ATBD-NO2-data-products, [Online; accessed 18-August-2021], 2021.

**Response to comment of Anonymous Referee #2 on "Cancellation of cloud shadow effects in the absorbing aerosol index retrieval algorithm of TROPOMI" by Victor Trees et al.**

Victor J. H. Trees [1,2], Ping Wang [1], Piet Stammes [1], Lieuwe G. Tilstra [1], David P. Donovan [1,2], and A. Pier Siebesma [2]

[1]Research & Development Satellite Observations, Royal Netherlands Meteorological Institute (KNMI), Utrechtseweg 297, 3731 GA, De Bilt, the Netherlands
[2]Department of Geoscience & Remote Sensing, Delft University of Technology, Stevinweg 1, 2628 CN, Delft, the Netherlands

**Correspondence:** Victor Trees (victor.trees@knmi.nl)

*Summary*

*In this paper the authors discuss the effect of cloud shadows in the magnitude of the qualitative Absorbing Aerosol Index (AAI) parameter derived from TROPOMI near UV observations and verify their findings with 3D radiative transfer (RT) calculations. The reported RT results are a good validation of their developed 3-D RT tool application to UV observations. They conclude*
*that the effect on the AAI is quite small and that, therefore, a correction is not necessary.*

*General Comments*

*The detection of cloud shadows signal in the AAI is not a new finding. Cloud shadows in equivalent AAI definitions have been previously identified in satellite observations by sensors with UV observational capability at sub-kilometer spatial resolution*
*[Fukuda et al., 2013; Tanada et al., 2023; Gogoi et al, 2023]. As in the previously published analyses, this manuscript concludes that the cloud shadow effect on the AAI is quite small and does not warrant a correction.*

We thank the reviewer for pointing our those references. Although cloud shadows are indeed discussed in Fukuda et al. (2013) and Gogoi et al. (2023), they do not discuss the cloud shadow effect on the absorbing aerosol index (AAI) in its form used by satellite spectrometers such as GOME, OMI and TROPOMI. The Cloud and Aerosol Imager (TANSO-CAI) is an imager measuring in bands, and Fukuda et al. (2013) and Gogoi et al. (2023) retrieved the aerosol optical thickness (AOT) and single
scattering albedo (SSA). We are not aware of scientific publications that discussed cloud shadow effects on the AAI. Therefore, we consider our analysis of cloud shadow effects on the TROPOMI AAI as novel.

*This conclusion can be further justified by the fact that the AAI is not a physically meaningful parameter. Thus, in its current*
*form, the manuscript's main contribution is the validation of the MONKI UV radiative transfer scheme and could, perhaps, be submitted to an RT specialized journal.*

Although the AAI is not a quantity with physical units, features in AAI maps can sometimes be related to other physical phenomena than absorbing aerosols. For example, Kooreman et al. (2020) discuss the AAI increase due to the cloud bow. In our paper, we study the impact of cloud shadows. As we discuss in the introduction of the paper, AAI features that are not directly related to absorbing aerosols require an explanation and/or correction in order to guarantee the quality of this satellite product. Therefore, we think AMT is the appropriate journal for this work.

*The authors could advantageously use the documented AAI-cloud shadow detection capability to develop corrections to radiance measurements at relevant channels in the UV-VIS-near-IR range that are used in the retrieval of important geophysical parameters. Although corrections would not be necessary for DOAS (Differential Optical Absorption Spectroscopy) based applications, they would be needed for cloud and aerosol algorithms and other inversion schemes based on the interpretation of discrete channel observations. One such AAI-based reflectance correction was developed and applied to GOSAT/TANSO CAI observations at 672 nm (Fukuda et al., 2013). Similar applications to TROPOMI measured radiances would be an important contribution to improve the accuracy of retrieved physical parameters and contribute towards the error budget analyses of TROPOMI retrieved parameters. I encourage the authors to pursue this goal.*

The thank the reviewer for this suggestion. We indeed have the cloud shadow detection DARCLOS which can be used to identify cloud shadow pixels. Because in this paper we conclude that the average cloud shadow effect on the AAI is not apparent, we think that the AAI itself is not a suitable parameter to be part of a potential correction strategy of radiances in the cloud shadow for other TROPOMI products such as the aerosol optical thickness (AOT) (de Graaf, 2022). In addition, we think that first the cloud shadow impact on the AOT should be analyzed (as we did for this paper for the AAI), before efforts are made to design a correction strategy.

*Specific Comments*

*Line 19: Suggest replacing spectrometer with hyperspectral instrument.*

We thank the reviewer for the suggestion. The instrument type name of TROPOMI is commonly written as 'spectrometer', see e.g. the TROPOMI paper (Veefkind et al., 2012) or the KNMI and SRON websites: https://www.knmi.nl/kennis-en-datacentrum/project/tropospheric-monitoring-instrument-tropomi and https://earth.sron.nl/project/tropomi/. Therefore, we keep the sentence as is.

*Line 22: Suggest adding hyperspectral before predecessors.*

We think that adding the word 'hyperspectral' in this sentence would be redundant, and therefore confusing, since the word 'its' in this sentence already refers to predecessors of TROPOMI (which is a spectrometer). Therefore, we keep the sentence as is.

*Line 24: It can be said that TROPOMI's 3.6X5.6 km spatial resolution is indeed unprecedented in hyperspectral sensors. However, that is not the case for multi-wavelength sensors. Near UV channels at sub-kilometer resolution have been added to*

*several sensors over the last decade. In 2009, the Cloud Aerosol Imager (CAI) sensor on the Japanese GOSAT TANSO satellite made 380 nm radiance measurement at 0.25 km resolution (Fukuda et al., 2013). Similar measurements were carried out by*

*the Second-generation Global Imager (SGLI) launched in 2017 (Tanada et al, 2023). Measurements at 339 and 377 nm at 0.46 km resolution have been made by CAI-2 since its launch in 2018 (Gogoi et al., 2023). The recently deployed PACE-OCI instrument measures UV radiances at 1 km spatial resolution (Werdell et al., 2019).*

Our paper focusses on the signals in hyperspectral sensors (at high spectral resolution) rather than multi-wavelength imagers (which use wavelength bands). For the AAI, two wavelengths in the UV (sampled at high spectral resolution) are used. Hence, we have not included a discussion of the measurements taken by imagers in our paper.

*Lines 34-35: Another important non-aerosol related effect on measured near UV radiances is pure water absorption over the remote oceans. Suggest adding 'as well as pure water absorption effects over the remote oceans (Fry, 2000)' after (Kooreman et al., 2020).*

Possible spectral effects of dissolved matter and chlorophyll in the ocean could indeed affect the AAI. We have added 'constituents in the ocean' as follows:

Lines 34-35: *Hence, AAI features that are not related to absorbing aerosols, for example caused by the ocean glint, absorbing constituents in the ocean water, and clouds at specific scattering geometries (Kooreman et al., 2020), may be undesired for those retrievals.*

*Line 39: Suggest adding hyperspectral before predecessors.*

We leave the sentence as is. See previous answer to comment regarding line 22, above.

*Line 40: A brief discussion on earlier studies on cloud shadows in the near UV applied to observations by other high spatial*

*resolution multi-wavelength instruments (JAXA's CAI, SGLI and CAI-2) should be included here.*

We do not believe a discussion on cloud shadows in imager data would improve the introduction of the paper. We think that the reader could be confused by such a discussion, due to the different type of instruments and different type of satellite products.

*Line 99. Suggest qualifying or removing the statement 'In the absence of aerosols and clouds, the AAI is, in theory, ideally*

*equal to zero'. Actually, there are multiple non-aerosol related effects such as land surface reflectance spectral effects (deserts in particular) and ocean signal associated with sunglint and clear water absorption (both yielding positive AAI values) as well as chlorophyll absorption that yields negative AAI values. An explanation of how these non-aerosol related effects are detected and flagged should be included.*

Those other AAI-modifying phenomena indeed could, in principle, contaminate our data set with shadow-pixels and non- shadow pixels. However, we think that there is no reason to believe that the probability to, for example, encounter chlorophyll is higher in a shadow pixel than in shadow-free neighbour pixel. Since our analysis focuses on the differences between the shadow- and shadow-free neighbour pixels, we do not expect those phenomena to change the conclusions of our paper. We consider those effects to be part of the 'natural variation irrespective of cloud shadows', which is captured by the comparison of the first and second shadow-free neighbour pixels (see Sect. 3.2).

In the introduction we already mentioned that '*[...], AAI features that are not related to absorbing aerosols, for example caused by the ocean glint, absorbing constituents in the ocean water, and clouds at specific scattering geometries (Kooreman et al., 2020), may be undesired [...]*'. To further clarify the sentence on line 99, we have modified it as follows:

Line 99:  –> *In a scene without aerosols and*

*clouds, above a spectrally neutral Lambertian surface, the AAI is, in theory, equal to zero.*

*Line 100 Elaborate on the reasons for such a large negative offset. Is the reported AAI adjusted for this offset? If so, how? Provide references on this issue.*

The AAI bias was not corrected for in this paper. We have added the following sentences to line 100:

*The offset in the collection 1 AAI data used for this paper is due to radiometric calibration offsets and degradation in the TROPOMI radiance and irradiance data (Tilstra et al., 2020; Ludewig et al., 2020). The degradation in the radiance and irradiance results in an increase in the derived reflectances at 340 and 380 nm, decreasing the average AAI values over time.*

*Line 101 On the detectability of aerosols above clouds suggest adding the Torres et al (2012) reference.*

We thank the reviewer for the suggestion. We have added this reference to line 101.

References

Fry, E.S., Visible and near ultraviolet absorption spectrum of liquid water. Appl. Opt. 2000, 39, 2743–274

Fukuda, S., T. Nakajima, H. Takenaka, A. Higurashi, N. Kikuchi, T. Y. Nakajima, and H. Ishida (2013), New approaches to removing cloud shadows and evaluating the 380nm surface reflectance for improved aerosol optical thickness retrievals from the GOSAT/TANSO-Cloud and Aerosol Imager, J. Geophys. Res. Atmos., 118, 13,520–13,531, doi:10.1002/2013JD020090.

Gogoi, M. M., Babu, S. S., Imasu, R., and Hashimoto, M.: Satellite (GOSAT-2 CAI-2) retrieval and surface (ARFINET) obser-
vations of aerosol black carbon over India, Atmos. Chem. Phys., 23, 8059–8079, https://doi.org/10.5194/acp-23-8059-2023, 2023.

Tanada, K., Murakami, H., Hayasaka, T., & Yoshida, M. (2023). Aerosol optical properties of extreme global wildfires and estimated radiative forcing with GCOM-C SGLI. Journal of Geophysical Research: Atmospheres, 128, e2022JD037914.
https://doi.org/10.1029/2022JD037914

Torres, O, H. Jethva, and P.K. Bhartia, Retrieval of Aerosol Optical Depth above Clouds from OMI Sensitivity Analysis and Case Studies, Journal. Atm. Sci., 69, 1037-1053 Observations: Sensitivity Analysis and Case Studies, Journal. Atm. Sci., 69, 1037-1053, doi:10.1175/JAS-D-11-0130.1, 2012

Werdell, P.J., Behrenfeld, M.J., Bontempi, P.S., Boss, E., Cairns, B., Davis, G.T., Franz, B.A., Gliese, U.B., Gorman, E.T., Hasekamp, O., Knobelspiesse, K.D., Mannino, A., Martins, J.V., McClain, C.R., Meister, G., and Remer, L.A. (2019). The Plankton, Aerosol, Cloud, Ocean Ecosystem Mission: Status, Science,

**References**

de Graaf, M.: TROPOMI ATBD of the Aerosol Optical Thickness. Doc. No. S5P-KNMI-L2-0033-RP, Issue 3.0.0, Royal Netherlands Meteorological Institute (KNMI), https://data-portal.s5p-pal.com/product-docs/aot/s5p_aot_atbd_v3.0.0_2022-02-10_signed.pdf, [Online; accessed 18-January-2024], 2022.

Kooreman, M. L., Stammes, P., Trees, V., Sneep, M., Tilstra, L. G., de Graaf, M., Stein Zweers, D. C., Wang, P., Tuinder, O. N. E., and Veefkind, J. P.: Effects of clouds on the UV Absorbing Aerosol Index from TROPOMI, Atmospheric Measurement Techniques, 13,

6407–6426, https://doi.org/10.5194/amt-13-6407-2020, 2020.

Ludewig, A., Kleipool, Q., Bartstra, R., Landzaat, R., Leloux, J., Loots, E., Meijering, P., van der Plas, E., Rozemeijer, N., Vonk, F., and Veefkind, P.: In-flight calibration results of the TROPOMI payload on board the Sentinel-5 Precursor satellite, Atmospheric Measurement Techniques, 13, 3561–3580, https://doi.org/https://doi.org/10.5194/amt-13-3561-2020, 2020.

Tilstra, L. G., de Graaf, M., Wang, P., and Stammes, P.: In-orbit Earth reflectance validation of TROPOMI on board the Sentinel-5 Precursor satellite, Atmospheric Measurement Techniques, 13, 4479–4497, https://doi.org/10.5194/amt-13-4479-2020, 2020.

Veefkind, J. P., Aben, I., McMullan, K., Förster, H., de Vries, J., Otter, G., Claas, J., Eskes, H. J., de Haan, J. F., Kleipool, Q., van Weele, M., Hasekamp, O., Hoogeveen, R., Landgraf, J., Snel, R., Tol, P., Ingmann, P., Voors, R., Kruizinga, B., Vink, R., Visser, H., and Levelt, P. F.: TROPOMI on the ESA Sentinel-5 Precursor: A GMES mission for global observations of the atmospheric composition for climate, air quality and ozone layer applications, Remote Sensing of Environment, 120, 70–83, https://doi.org/10.1016/j.rse.2011.09.027, 2012.